# Amygdala-ventral striatum circuit activation decreases long-term fear

Susana S Correia[1,2]*, Anna G McGrath[1,2], Allison Lee[1,2], Ann M Graybiel[1,2], Ki A Goosens[1,2]*

[1]McGovern Institute for Brain Research, Massachusetts Institute of Technology, Cambridge, United States; [2]Department of Brain and Cognitive Sciences, Massachusetts Institute of Technology, Cambridge, United States

**Abstract** In humans, activation of the ventral striatum, a region associated with reward processing, is associated with the extinction of fear, a goal in the treatment of fear-related disorders. This evidence suggests that extinction of aversive memories engages reward-related circuits, but a causal relationship between activity in a reward circuit and fear extinction has not been demonstrated. Here, we identify a basolateral amygdala (BLA)-ventral striatum (NAc) pathway that is activated by extinction training. Enhanced recruitment of this circuit during extinction learning, either by pairing reward with fear extinction training or by optogenetic stimulation of this circuit during fear extinction, reduces the return of fear that normally follows extinction training. Our findings thus identify a specific BLA-NAc reward circuit that can regulate the persistence of fear extinction and point toward a potential therapeutic target for disorders in which the return of fear following extinction therapy is an obstacle to treatment.

*For correspondence: scorreia@ mit.edu (SSC); kgoosens@mit.edu (KAG)

**Competing interests:** The authors declare that no competing interests exist.

## Introduction

Anxiety, trauma, and stress-related disorders have a high lifetime prevalence rate (*Kessler et al., 2005*) and can be debilitating. The most common treatment for these disorders is exposure therapy, in which trauma-associated or anxiety-provoking cues are presented in a safe environment in order to decrease gradually, or to extinguish, cue-evoked recollection (*Rothbaum and Davis, 2003*; *Felmingham et al., 2013*). A challenge in treating these disorders, however, is the re-emergence of symptoms following therapy, a phenomenon called spontaneous recovery (*Pavlov, 1927*; *Rescorla, 2004*). The existence of spontaneous recovery suggests that aversive memories associated with the traumatic incident are temporarily inhibited, but not erased, by exposure therapy and that extinction memories are weakened by the passage of time. Understanding how to strengthen extinction memories so that fear return is lessened is thus critical in designing improved treatment of many fear-related and anxiety disorders.

In the laboratory, traumatic events and exposure therapy are modeled using Pavlovian fear conditioning and extinction training. During fear conditioning, neutral cues (conditional stimuli, CSs) are paired with aversive cues (unconditional stimuli, USs), to induce learned fear. During extinction training, the CSs are presented in the absence of aversive stimuli in order to decrease CS-elicited fear. Associative fear memories that result from fear conditioning can be extinguished, but they exhibit spontaneous recovery (*Quirk, 2002*), providing a powerful behavioral model by which to identify and study the neural circuits that support these phenomena.

The nucleus accumbens (NAc), a striatal region classically linked to rewarding experiences (*Ambroggi et al., 2008*; *Ghitza et al., 2003*; *Wan and Peoples, 2006*), is thought also to participate in aversive memory (*Cassaday et al., 2005*; *Haralambous and Westbrook, 1999*; *Jongen-Rêlo et al., 2003*; *Riedel et al., 1997*; *Schwienbacher et al., 2004*; *Reynolds and Berridge, 2008*).

Numerous studies have shown that activity in the NAc is correlated with several measures of learned aversive behaviors (*Reynolds and Berridge, 2008*; *Delgado et al., 2009*; *Kim et al., 2006a*). Moreover, the NAc receives strong glutamatergic inputs from the basolateral amygdala (BLA) (*Ambroggi et al., 2008*; *Brog et al., 1993*; *Christie et al., 1987*; *Floresco et al., 2001*; *O'Donnell and Grace, 1995*; *Wright et al., 1996*), a brain region necessary for the acquisition and consolidation of fear extinction (*Herry et al., 2008*; *Laurent et al., 2008*) but not for retrieval or expression of the fear extinction memory (*Herry et al., 2008*). It has been suggested, accordingly, that activity in the NAc is important for fear extinction (*Holtzman-Assif et al., 2010*; *Whittle et al., 2013*; *Rodriguez-Romaguera et al., 2012*). The mechanisms underlying such a function of the NAc remain unclear, however, and it is not known whether the NAc is implicated in the persistence of extinction memory, a critical therapeutic question. Based on evidence that the BLA is essential for many forms of motivated learning, including fear extinction and reward learning (*Holtzman-Assif et al., 2010*; *LeDoux et al., 1990*; *Falls et al., 1992*; *Herry et al., 2006*; *Sotres-Bayon et al., 2007*; *Schoenbaum et al., 1998*; *Sugase-Miyamoto and Richmond, 2007*; *Kim et al., 2006b*; *Quirk et al., 1995*; *Bouton, 2002*; *Kelley et al., 2009*; *Gold et al., 2012*; *Belova et al., 2007*; *Paton et al., 2006*; *Uwano et al., 1995*; *Tye et al., 2008*; *Britt et al., 2012*; *Stuber et al., 2011*), and that connections between the BLA and the NAc are important for generating goal-directed behavior in response to reward-predictive cues (*Ambroggi et al., 2008*; *Wright et al., 1996*; *Britt et al., 2012*; *Stuber et al., 2011*; *Setlow et al., 2002*; *McDonald, 1991a*; *Di Ciano and Everitt, 2004*; *Johnson et al., 1994*), we here asked whether fear extinction itself engages a BLA-NAc circuit in rodents and, if so, whether enhancing activity in this circuit during extinction training could reduce the return of fear at a remote time point.

## Results

### A BLA-NAc circuit is recruited by fear extinction

To test whether a BLA-NAc circuit is engaged by fear extinction, we first used a double-labeling strategy to search for NAc-projecting BLA neurons that could be activated by extinction training after fear conditioning. We labeled BLA neurons projecting to the NAc by infusing a fluorescent retrograde tracer, cholera toxin B (CTB) conjugated to Alexa fluor 488 (*Conte et al., 2009*), into the NAc to allow subsequent detection of labeled BLA-NAc projecting neurons (*Figure 1*). Approximately two weeks after surgery, rats were subjected to auditory fear conditioning followed by either (1) a brief auditory fear recall session (*Fear recall* group; *Figure 1A*) or (2) two sessions of auditory fear extinction training (*Long ext* group; *Figure 1A*), which promoted significant extinction learning and retention (*Figure 1—figure supplement 1A–B*). Three control groups were prepared as follows. In a *Naïve* group, rats were kept in their home cage (*Figure 1A*). In a *Context* group, rats were given standard auditory fear conditioning followed by two sessions of exposure to a novel context (*Figure 1A*). In a *Tone* group, rats were exposed to the same number of tones and context exposure as the *Long ext* group, but without experiencing the footshock US on Day 1 (*Figure 1A*).

Comparisons of the behavior of the rats on Day 2 (*Figure 1—figure supplement 1A*) showed that rats in the *Fear recall* and *Long ext* groups showed significantly higher levels of freezing than rats in the *Context* and *Tone* groups. Further, the *Long ext* group displayed substantial across-session extinction memory on the second session of extinction training (*Figure 1—figure supplement 1B*). The rats from all groups were perfused one hour after the final behavioral session, a time point corresponding to heightened cFos expression (*Figure 1C*, *Figure 1—figure supplement 1D*), and immunohistochemistry for cFos was used to identify active BLA neurons (cFos+, *Figure 1B*). In the *Fear recall* group, this labeled the BLA neurons activated by fear memory recall, whereas in the *Long ext* group, this labeled the BLA neurons activated by fear memory recall, extinction memory recall, and additional fear extinction learning.

The rats subjected to fear conditioning and a fear recall test (*Fear recall* group) or fear conditioning and extinction training (*Long ext* group) exhibited far greater numbers of cFos+ BLA neurons than the control rats (*Figure 1C*). Furthermore, significantly greater numbers of cFos+ neurons were observed in the BLA after fear extinction training (*Ext recall* group) than in the *Fear recall* group after auditory recall test (*Figure 1C*).

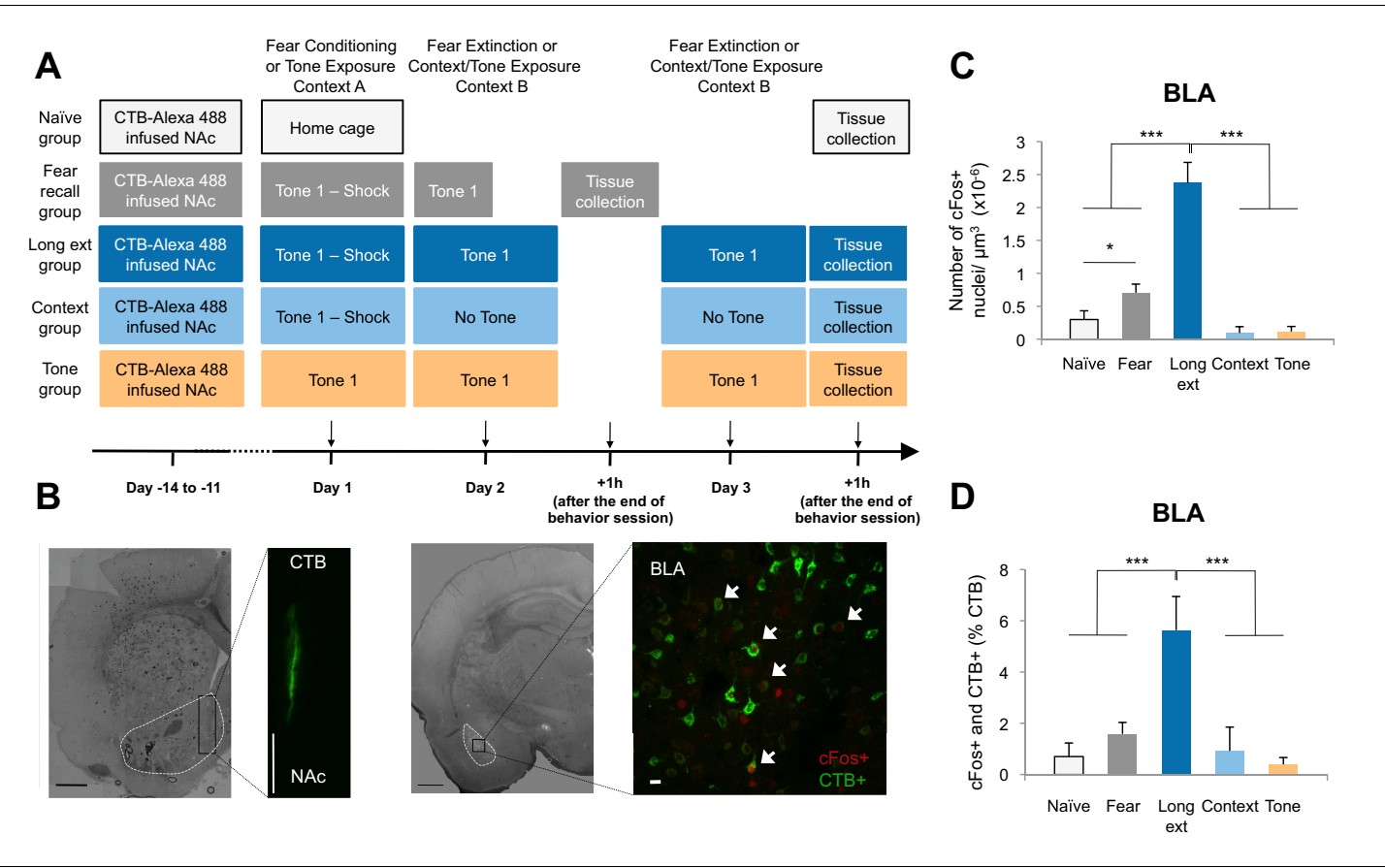

**Figure 1.** The BLA-NAc circuitry is recruited during extinction of fear. (A) Experimental design. (B) Coronal slices encompassing the BLA were stained for cFos protein and imaged, and the numbers of CTB+ cells (right panel in green), cFos+ nuclei (right panel in red) and double-labeled cells (right panel, white arrows indicate a subset of this population) were counted (*Fear recall* group: n = 21 from 3 rats; *Long ext* group: n = 23 from 4 rats; *Context* group: n = 12 from 4 rats; *Tone* group, n = 10 from 4 rats; *Naïve* group, n = 14 from 3 rats). The right side panel depicts an overlapping image of a Z-projection for each imaged channel. Scale bar on right panel indicates 10 μm; all other scale bars indicate 1 mm. (C) Number of cFos+ cells per μm$^3$ in the BLA (main effect of group, p<0.0001). (D) BLA cells double-labeled for CTB and cFos, normalized to the total number of CTB+ cells (main effect of group, p<0.0001). 'n' represents the number of images analyzed. Statistical significance was calculated with Kruskal-Wallis followed by unpaired planned comparisons. All data are mean ± s.e.m. *p<0.05, ***p<0.001.

The following figure supplements are available for figure 1:

**Figure supplement 1.** Freezing behavior and retrograde labeling in the BLA.

**Figure supplement 2.** The BLA-NAc circuitry is recruited during fear conditioning and extinction of fear.

A subpopulation of the cFos+ neurons were also retrogradely labeled from the NAc, identifying them as BLA neurons projecting to the NAc (CTB+) and activated by training (cFos+; *Figure 1B* right image; white arrows indicate double-labeled neurons). This population of double-labeled BLA neurons, as a percentage of all CTB+ neurons, was highly enriched in the rats that experienced extinction training (*Figure 1D*) relative to the numbers in all other experimental groups, in which we found only low levels of double-labeled neurons despite comparable densities of retrogradely labeled (CTB+) BLA neurons (*Figure 1—figure supplement 1C*). The double-labeled population of cells, although small relative to the total population of NAc-projecting BLA neurons (~5%), represented a substantial portion (31%) of the total cFos+ population in the BLA following extinction training. Consistent with previous studies showing that the BLA-NAc projection is heavy (*McDonald, 1991a*), in the BLA sections that we analyzed, 26.68 ± 1.81% of all BLA cells projected to the NAc. Thus, 5% of this significant projection is a large number of individual cells.

These data suggest that fear extinction produces stronger activation of the BLA, including NAc-projecting neurons, than fear expression. However, it is also possible that the lower cFos+ numbers found in the *Fear recall* group compared to the *Long ext* group were related to the length of the behavior session (7.8 min versus 31.1 min). To examine this, we tested an additional group of rats in which CTB was infused into the NAc; the rats were later subjected to auditory fear conditioning followed by only one session of fear extinction (*Short ext* group, 31.1 min; *Figure 1—figure supplement 2A*). The levels of freezing during the first extinction session (*Figure 1—figure supplement 2B*) and the density of cFos+ nuclei in the BLA (*Figure 1—figure supplement 2C*) were similar between rats in the *Short ext* and *Long ext* groups. However, the number of double-labeled cells was significantly higher after two extinction sessions (*Long ext* group) compared to one extinction session (*Short ext* group) (*Figure 1—figure supplement 2D*).

To determine whether the activation of the BLA-NAc pathway was specific to fear extinction, we ran an additional group of rats in which CTB was infused into the NAc. The rats were later exposed to auditory fear conditioning only (*Fear cond* group; *Figure 1—figure supplement 2A*). The rats in the *Fear cond* group displayed significantly higher numbers of cFos+ BLA neurons (*Figure 1—figure supplement 2C*) than rats in the *Short ext* and *Long ext* groups. Interestingly, the level of double-labeled cFos+ and CTB+ cells was not significantly different between the *Fear cond* and the *Long ext* groups (*Figure 1—figure supplement 2D*), despite the higher numbers of cFos+ BLA cells in the *Fear cond* group (*Figure 1—figure supplement 2C*).

This result raises the possibility that the BLA-NAc projection might contain a specialized circuit by which the ventral striatum can become activated during both fear conditioning and fear extinction learning. This possibility is supported by a strong negative correlation between activation of the BLA-NAc pathway and the strength of freezing during the extinction session (*Figure 1—figure supplement 2F*): greater recruitment of this pathway was associated with weaker fear memories.

## Reward conditioning during fear extinction reduces the return of fear

Our finding that a BLA-NAc circuit is recruited by fear extinction (*Figure 1*), together with evidence that a BLA-NAc circuit is involved in reward-seeking (*Britt et al., 2012*; *Stuber et al., 2011*), suggested that reward-related activity in the NAc during fear extinction could reinforce extinction learning. We therefore tested whether reward conditioning, a manipulation that activates NAc neurons (*Wan and Peoples, 2006*; *Day et al., 2006*), would produce stronger extinction memory than fear extinction training alone (*Figure 2A*).

This procedure, in which a predictive cue becomes associated with an outcome of opposing valence, is termed counterconditioning. In patients with fear and anxiety disorders, exposure therapy is regularly administered by exposing patients to aversive cues in the absence of aversive outcomes and in limited instances, counterconditioning has been used to treat post-traumatic stress disorder (PTSD) by pairing fear-eliciting stimuli with the recollection of positive memories during exposure therapy (*Paunović, 2003*; *Wolpe, 1958*; *Paunović, 2011*). However the mechanism by which counterconditioning achieves therapeutic efficacy is completely unknown. We therefore tested the effects of counterconditioning not only on immediate fear recall but also on fear recall after a delay of nearly two months.

We trained rats with auditory fear conditioning and then followed the initial training either with exposure to four sessions of auditory fear extinction (*Ext-Ext* group), or with exposure to two sessions of auditory fear extinction followed by two sessions of fear extinction and discriminative reward conditioning (*Ext-RC* group). The two sessions of auditory fear extinction that preceded the two sessions of discriminative reward conditioning in the *Ext-RC* group allowed for the reduction of fear expression and initiation of exploratory behavior necessary for the acquisition of reward conditioning. During the discriminative reward conditioning, the tone that was originally paired with footshock was paired with sucrose delivery at a port on the chamber wall (*Figure 2A*), and, as a control, a second novel tone was presented without sucrose delivery. We assessed tone-elicited freezing behavior and reward-seeking behavior (nose-pokes in the reward port) both during reward conditioning and also 55 days after fear conditioning. This paradigm enabled us to determine the effect of reward conditioning during fear extinction on the persistence of long-term extinction memory, a critical yet unexplored assessment when considering the translational significance of counterconditioning in humans with anxiety and trauma-related disorders.

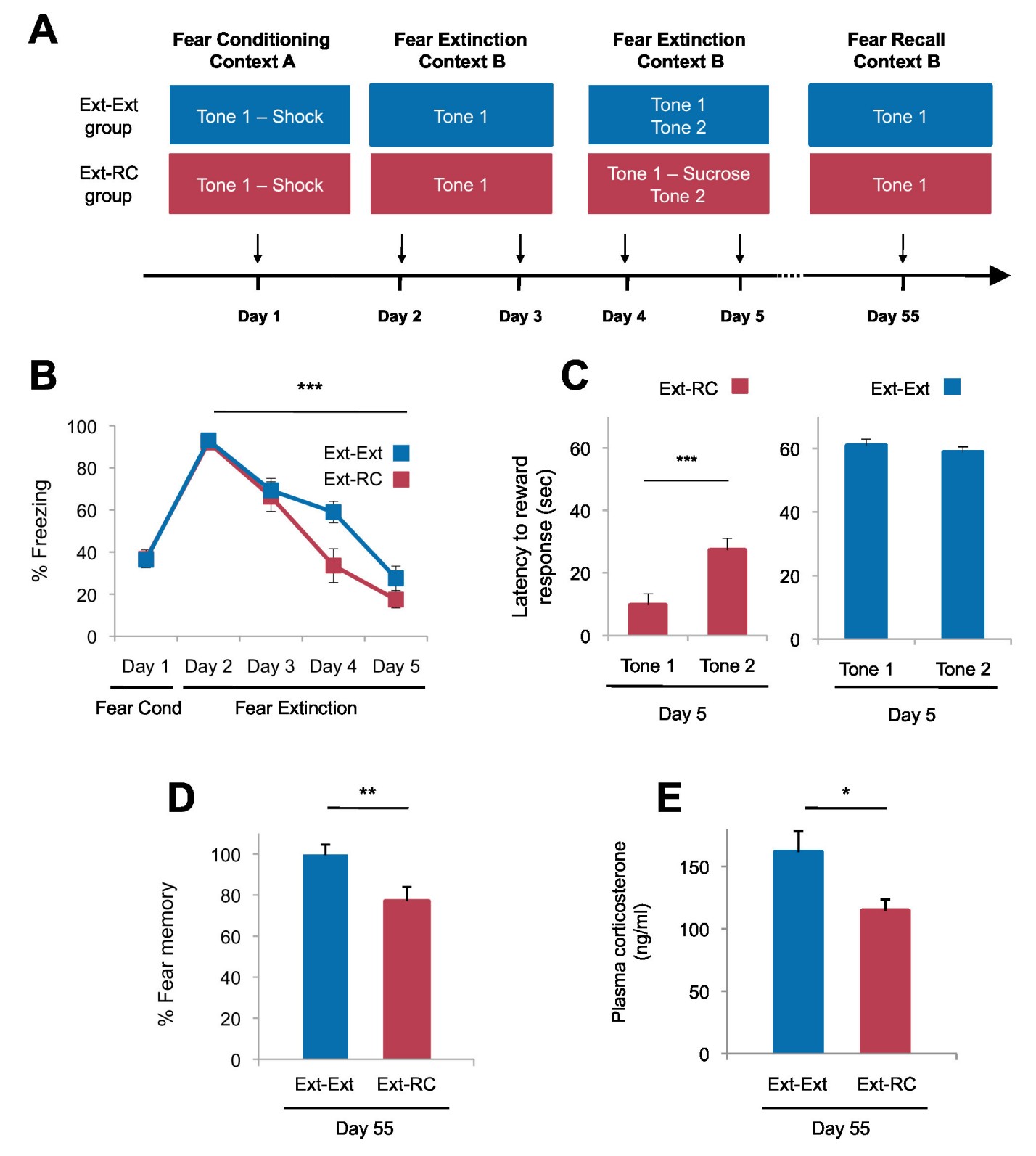

**Figure 2.** Tone-reward pairing during auditory fear extinction impedes the return of fear. (**A**) Experimental design (*Ext-Ext* group, n = 18; *Ext-RC* group, n = 16). (**B**) Fear to the tone was measured as the percent of time spent freezing during Tone 1 trials within the first five trials of each behavioral session (one session per day, as indicated; main effect of group, p=*n.s.*; for Days 2 to 5: main effect of day, p<0.0001). (**C**) Reward learning was assessed by the latency to respond at the reward port after Tone 1 (reward paired tone for group *Ext-RC*, n = 16) and Tone 2 (neutral tone), for rats in the *Ext-RC* group
*Figure 2 continued on next page*

*Figure 2 continued*

(p<0.001) and rats in the *Ext-Ext* group (n = 18, p=*n.s.*). (D) Return of fear on Day 55 was calculated as the percent of time spent freezing in the first five trials of Tone 1 presentations of Day 55 normalized (per rat) to the percent of time spent freezing during the first five trials of Tone 1 presentations of Day 2 (p<0.01). (E) Plasma corticosterone levels from rats in the *Ext-Ext* and *Ext-RC* groups on Day 55 (n = 12 for *Ext-Ext*, n = 9 for *Ext-RC*; p<0.05). 'n' represents the number of animals. Statistical significance was calculated using the Wilcoxon paired test (C) or two-tail Mann-Whitney test (B,D,E) followed by planned paired comparisons (B). All data are mean ± s.e.m. *p<0.05, **p<0.01, ***p<0.001.

The following figure supplements are available for figure 2:

**Figure supplement 1.** Tone-reward pairing during auditory fear extinction impairs return of fear.

**Figure supplement 2.** Fear expression is inhibited when reward is delivered during the recovery test.

**Figure supplement 3.** Fear return is not affected by reward conditioning when the fear-associated and reward-associated tones are different.

Rats in the two groups displayed comparable levels of tone-elicited freezing during fear conditioning and auditory fear extinction training (*Figure 2B*). On Days 4 and 5, only the rats that received reward conditioning (*Ext-RC* group) displayed discriminative tone-elicited reward responses (*Figure 2C*, *Figure 2—figure supplement 1A*). Long-term fear memory, measured 55 days after fear conditioning, was significantly lower in the *Ext-RC* group compared to the *Ext-Ext* group (*Figure 2D*, *Figure 2—figure supplement 1B–D*), despite the low overall number of reward port entries during the fear recall test on Day 55 (*Figure 2—figure supplement 1E*). Plasma corticosterone levels were also lower in the *Ext-RC group*, as compared to the *Ext-Ext* group, on Day 55 (*Figure 2E*), suggesting that the reduced fear recall on Day 55 in the *Ext-RC* group was accompanied by lower stress levels (*Kelley et al., 2009*).

The lower freezing observed in the *Ext-RC* group did not simply reflect reward-seeking behavior that masked conditional freezing. During the fear recall test on Day 55, freezing in the time bins during which a port entry occurred was not significantly different from freezing during time bins without port entries (*Figure 2—figure supplement 1F*). There was also no correlation between the number of reward port entries and the amount of freezing displayed (*Figure 2—figure supplement 1G*).

The lower freezing observed in the *Ext-RC* group also could not be attributed to consumption of sucrose (*Figure 2—figure supplement 2*). We ran an additional experiment to determine whether the tone-reward association formed during fear extinction (counterconditioning) would be preserved after several weeks and whether having sucrose present during the remote fear recall test would promote a greater decrease in fear (*Ext-RC* Group; *Figure 2—figure supplement 2A*). We assessed tone-elicited freezing behavior and reward-seeking behavior (nose-pokes in the reward port) during reward conditioning and also 55 days after fear conditioning (*Figure 2—figure supplement 2A*). During this remote test, the extinguished tone was paired with sucrose delivery for both groups, and the return of the original fearful memory was measured.

Rats in the two groups displayed comparable levels of tone-elicited freezing during fear conditioning and auditory fear extinction training (*Figure 2—figure supplement 2B*; Days 1–3). Similar freezing levels across the two groups were also observed on Days 4 and 5 (data not shown). On Days 4 and 5, only rats that received reward conditioning (*Ext-RC* group) displayed discriminative tone-elicited nose-pokes (*Figure 2—figure supplement 2C–D*). Rats that received only auditory fear extinction showed significant fear memory on Day 55 (in blue, *Figure 2—figure supplement 2E–G*), but rats that received fear extinction followed by reward conditioning, surprisingly, exhibited virtually no conditional freezing (in magenta, *Figure 2—figure supplement 2E–G*). Notably, for rats in the *Ext-RC* group, the discriminative responding on the reward task on Day 55 was as robust as on Day 5 (*Figure 2—figure supplement 2H*).

The lower freezing observed in the *Ext-RC* group also could not be attributed to the acquisition of a new rewarding association *per se* (*Figure 2—figure supplement 3*). To test whether it is specifically the reversal of the valence associated with the predictive tone that is critical for enhancing the persistence of auditory extinction memory, we performed an additional control experiment in which rats were trained with auditory fear conditioning followed by two sessions of auditory fear extinction and two sessions of reward conditioning, similar to the training regimen for the *Ext-RC* group.

However, in contrast to the procedure for the *Ext-RC* group, during reward conditioning, the tone that was originally paired with footshock was presented in the absence of footshock, and a second, novel tone was paired with sucrose delivery (*Figure 2—figure supplement 3A*). Despite the successful discriminative reward learning in this group of rats (*Figure 2—figure supplement 3B*), the reward conditioning during extinction did not significantly change the return of fear when compared to rats that received only extinction training (*Figure 2—figure supplement 3C*). Thus simply consuming sucrose or forming a new reward association with a novel cue during fear extinction was not sufficient to enhance fear extinction memory.

These findings demonstrate (1) that shifting the associative value of a tone cue during extinction from aversive to rewarding is possible, and that the positive value can be preserved for many weeks, (2) that reward conditioning during fear extinction does not alter across-session acquisition of fear extinction, although it is possible that a potential enhancement of extinction learning could have been masked in our experiments by the overall low levels of freezing observed on Days 4 and 5 and (3) that enhancing activation of reward circuits during fear extinction via a behavioral counterconditioning manipulation leads to more persistent weakening of the original fear memory than fear extinction alone.

## Reward conditioning during fear extinction promotes recruitment of a BLA-NAc circuit

We next tested whether the persistent reduction of fear return that we observed in the *Ext-RC* rats could reflect enhanced recruitment of the BLA-NAc circuit resulting from the reward training during fear extinction. We infused CTB conjugated to Alexa fluor 488 into the NAc of rats to retrogradely label BLA neurons projecting to the NAc (*Figure 3A*), and, after 11 to 14 days of recovery, we administered auditory fear conditioning followed either by three sessions of auditory fear extinction (*Ext-Ext* group) or by two sessions of auditory fear extinction and one session of reward conditioning combined with fear extinction (*Ext-RC* group). We also tested a control group trained on one session of reward conditioning without any aversive conditioning (*RC* group). Brains were taken for cFos immunolabeling of the BLA one hour after the final training session, and the proportion of BLA neurons projecting to the NAc (CTB+) that were activated during extinction training (cFos+) was determined by fluorescence confocal microscopy.

Extinction during the days following fear conditioning (Days 2–4) was again similar between the *Ext-Ext* and *Ext-RC* groups (*Figure 3—figure supplement 1A*). Also as expected, rats in both the *Ext-RC* and *RC* groups, which were trained on reward conditioning, displayed reward-seeking behavior, but such behavior was not observed in the *Ext-Ext* group (*Figure 3—figure supplement 1B–D*). We found significantly more BLA neurons that project to the NAc [neurons double-labeled for cFos and CTB (as a percentage of CTB+ neurons)] in the *Ext-RC* group (*Figure 3B*) as compared to the *Ext-Ext* group, despite similar densities of CTB+ cells in the BLA in the two groups (*Figure 3—figure supplement 1E*). Notably, the proportion of BLA neurons double-labeled for cFos and CTB (*Figure 3B*, inset) were similar in the *Ext-RC* and the *RC* group that only received reward conditioning. Also, there was a strong trend for higher numbers of cFos+ neurons in the BLA in the *Ext-RC* group as compared to the *Ext-Ext* group (*Figure 3C*), but this did not reach statistical significance.

These results demonstrate that the activity of the BLA and the BLA-NAc circuit is increased by reward conditioning during fear extinction, and that this activity is comparable to the levels observed following reward conditioning alone. Moreover, our results also show that reward conditioning combined with fear extinction increases the proportion of activated BLA neurons that project to the NAc, relative to the levels observed following fear extinction training alone. These findings enhance previous findings showing that the BLA itself participates in the acquisition and consolidation of fear extinction (*Herry et al., 2008*; *Laurent et al., 2008*) by demonstrating that the activity of a BLA-NAc circuit is increased by reward conditioning during fear extinction acquisition.

To compare the levels of recruitment of NAc neurons during fear extinction combined with reward conditioning to recruitment levels during fear extinction alone, we performed immunostaining for the transcription factor, Nr4a3, to assess activity of neurons in the NAc (*Figure 3D*) (*Hawk and Abel, 2011*; *Werme et al., 2000*). Using microarray analysis, we found that Nr4a3, but not cFos, was rapidly induced in the NAc by tone-reward association (*Figure 3—figure supplement 2*). Additionally, cFos immunolabeling in the NAc of rats that experienced fear extinction and/or reward conditioning did not yield detectable immunostaining (not shown), in contrast with the

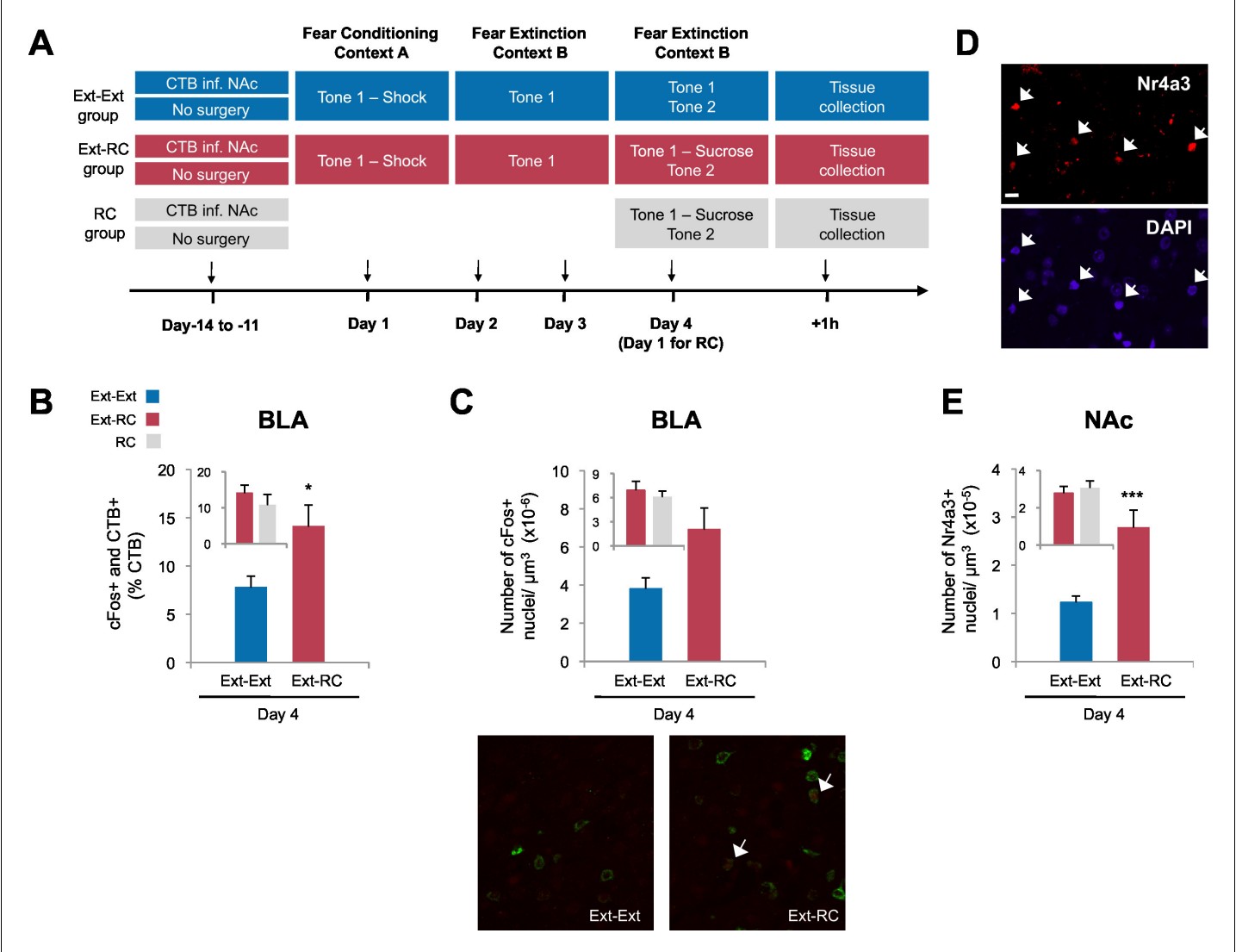

**Figure 3.** Recruitment of a BLA-NAc circuit is greater following fear extinction with reward conditioning than fear extinction alone. (**A**) Experimental design. Coronal slices encompassing the amygdala were stained for cFos protein and imaged, and numbers of CTB+ cells, cFos+ nuclei and double-labeled cells were counted (*Ext-Ext* group, B and C: n = 22 from 3 rats, E: n = 15 from 4 rats; *Ext-RC* group, B and C: n = 21 from 5 rats, E: n = 11 from 3 rats; *RC* group, B and C: n = 24 images from 3 rats, E: n = 19 images from 5 rats). (**B**) BLA cells double-labeled for CTB and cFos, normalized to the total of CTB+ cells (p<0.05; inset p=*n.s.*). (**C**) Number of cFos+ cells per μm³ in the BLA (top panel, p=0.07; inset p=*n.s.*). Representative images for data shown on panels B and C (bottom panel). (**D**) Coronal slices encompassing the NAc were stained for Nr4a3 protein and for nuclei (DAPI) and imaged, and numbers of Nr4a3+ nuclei were counted. A Z-projection of each imaged channel is represented. Scale bar represents 10 μm. (**E**) Number of Nr4a3+ cells per μm³, in the NAc (p<0.001; inset p=*n.s*). 'n' represents the number of images analyzed. Statistical significance was calculated with two-tail t-test for unpaired data (B, C inset) or two-tail Mann-Whitney test (B inset, C, E, E inset). All data are mean ± s.e.m. *p<0.05, ***p<0.001.

The following figure supplements are available for figure 3:

**Figure supplement 1.** Freezing and reward behavior and labeling densities in the BLA.

**Figure supplement 2.** Microarray analysis of the NAc after reward conditioning in rats.

**Figure supplement 3.** Density of Nr4a3 labeling in the NAc after behavioral training.

consistent immunostaining of cFos in other brain regions (*Figures 1*, *3*, and 6). Immunostaining (*Figure 3E*) showed significantly higher numbers of Nr4a3+ cells in the NAc of rats in the *Ext-RC* group than in the NAc of rats in the *Ext-Ext* group. Furthermore, both the *Ext-Ext* and the *Ext-RC* groups showed significantly higher density of Nr4a3+ cells in the NAc as compared to the *Naïve* group (*Figure 3—figure supplement 3*). Notably, reward conditioning combined with fear extinction and reward conditioning alone produced similar densities of Nr4a3+ nuclei (*Figure 3E*, inset). These results suggest that reward conditioning during fear extinction recruited the NAc more effectively than fear extinction training alone.

## Anatomically-selective activation of a BLA-NAc circuit during extinction learning reduces the return of fear

Our evidence that a BLA-NAc circuit is activated by fear extinction raised the possibility that this circuit plays a causal role in facilitating the acquisition of long-term fear extinction memory. We addressed this issue directly by pairing optogenetic stimulation of BLA terminals in the NAc with tone presentations during extinction training (Figure 5B). To establish the feasibility of this approach, we first infused either an adeno-associated virus (AAV; serotype 9) expressing channelrhodopsin fused to enhanced yellow fluorescent protein (eYFP; ChETA-eYFP) or, as a control, a similar AAV expressing eYFP alone into the amygdala. After one month to allow anterograde expression of the virus, we used confocal imaging to analyze the distribution of the ChETA-eYFP or eYFP proteins in the BLA and NAc. We found eYFP expressed broadly in both brain regions, indicating successful expression and transport of the protein (*Figure 4A*). To test whether ChETA-eYFP was expressed in the presynaptic terminals of BLA neurons projecting to the NAc, we double-labeled NAc sections for eYFP (*Figure 4B*, in green) and synaptophysin, a marker for presynaptic terminals (*Figure 4B*, in red). ChETA-eYFP expression exhibited overlap with synaptophysin staining (*Figure 4B*, white arrows). This result confirmed that BLA axons projecting to the NAc form direct synaptic connections with NAc neurons, and that the ChETA-eYFP protein was appropriately located to enable us to activate selectively in the NAc the terminals of axons originating in the BLA.

To determine whether stimulation of these presynaptic BLA terminals could activate NAc neurons, we implanted optical fibers above the NAc of rats expressing AAV-ChETA-eYFP or control AAV-eYFP virus in the amygdala. We recorded multiunit activity in the BLA and the NAc of anesthetized rats during 5 s periods before, during and after the laser stimulation period (473 nm light stimulation with 10–20 mW, 20 Hz, 5 ms pulses at 1 min intertrial intervals; *Figure 4C*).

In agreement with previous observations (*Britt et al., 2012*), this pattern of stimulation was sufficient to elicit an increase in neuronal activation in the NAc when AAV-ChETA-eYFP was expressed (*Figures 4D–E*), but not when the control AAV-eYFP, lacking opsin, was expressed (*Figure 4E*). This stimulation also increased the number of Nr4a3+ cells in the NAc, as compared to the numbers in rats that underwent fear extinction without stimulation (*AAV-ChETA-eYFP* vs. *Ext-Ext*, *Figure 3—figure supplement 1F*). This increase of Nr4a3+ cells in the NAc indicates that stimulation of the BLA terminals in the NAc resulted in increased activity of NAc neurons.

To determine whether the optogenetic stimulation of this BLA-NAc circuit could enhance fear extinction memories, we then paired the optogenetic stimulation with fear extinction, following the same timeline in which reward conditioning enhanced the effects of fear extinction (*Figure 5A*). We bilaterally infused into the amygdala an AAV (serotype 9) expressing either ChETA-eYFP or eYFP alone, and implanted two optic fibers bilaterally targeting the NAc of these rats (*Figure 5A*, *Figure 5—figure supplement 1B*). One month later, the rats received auditory fear conditioning followed by two sessions of auditory fear extinction training, and two additional auditory fear extinction sessions in which we administered light stimulation for 5 s after the termination of each tone (10–20 mW, 473 nm, 20 Hz, 5 ms pulses, 65 s interstimulation interval; *Figure 5A–B*). This pattern of stimulation, previously shown to promote self-stimulation in mice (*Stuber et al., 2011*), did not enhance place preference in our rats (*Figure 5—figure supplement 1A–B*), suggesting that it was not inherently rewarding in our experimental setting. However, we found that the stimulation protocol did promote reward-seeking (*Figure 5—figure supplement 1C–E*).

While there were many potential time points at which we could have investigated the impact of stimulation on fear extinction and its persistence, several factors led us to select the time point described. We chose the time for laser stimulation to be immediately after the tone offset in order to match the timing of sucrose consumption by the rats in the *Ext-RC* group (*Figure 2A*). We chose

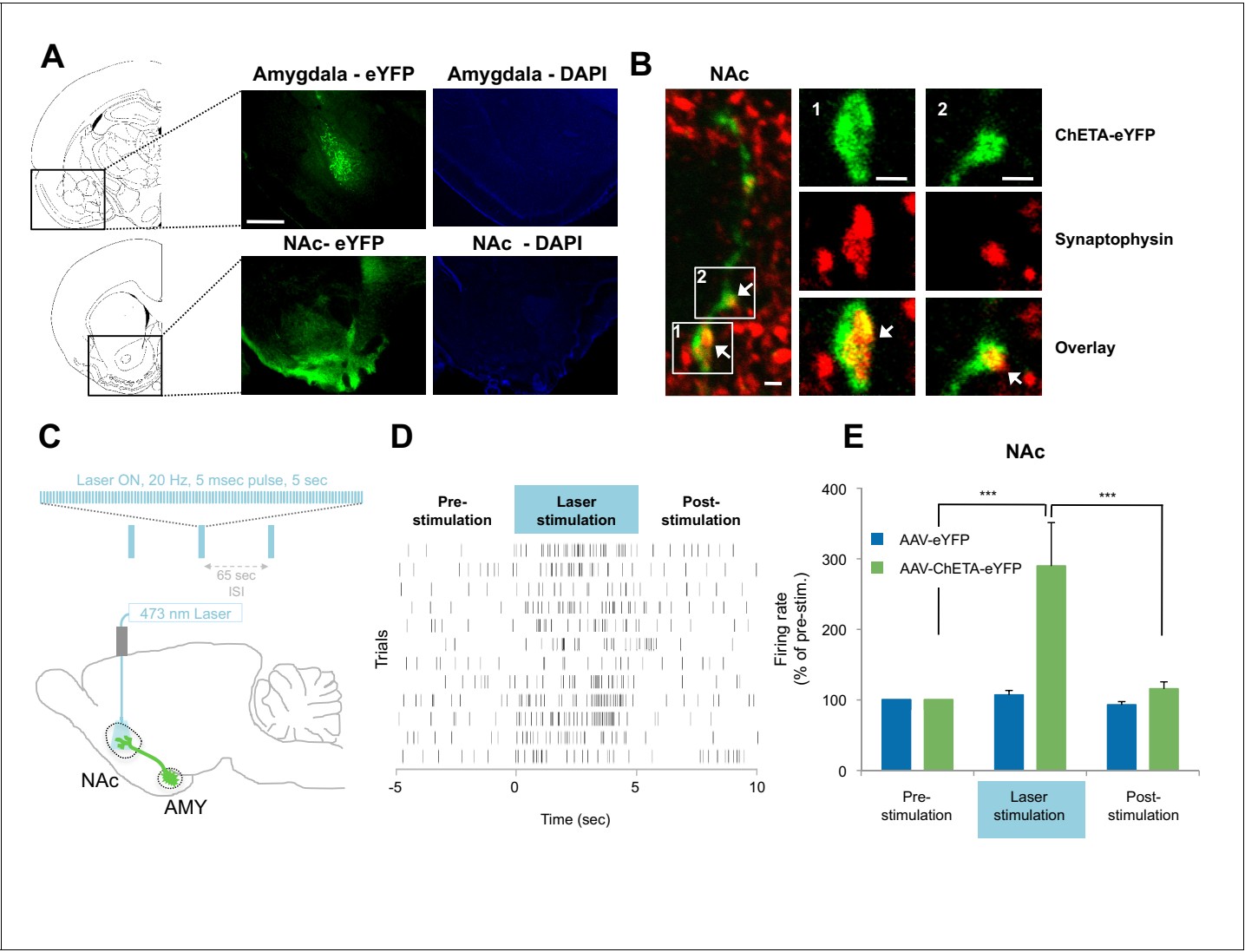

**Figure 4.** Optogenetic stimulation of amygdala presynaptic terminals projecting to NAc. (A) Expression of ChETA-eYFP (left) and DAPI (right) in the BLA and NAc of rats infused with AAV-ChETA-eYFP in the amygdala. Scale bar represents 1 mm. (B) Expression of ChETA-eYFP in presynaptic terminals in the NAc was confirmed by immunolabeling of ChETA-eYFP and synaptophysin, as indicated. White arrows indicate overlapping staining. Scale bars represent 1 μm. (C) Schematic representation of the light stimulation parameters used for *in vivo* recordings. (D, E) *In vivo* recordings from the NAc of anesthetized rats infused with AAV-ChETA-eYFP or AAV-eYFP in the amygdala and implanted with fiber optics above the NAc. Optic fibers delivered 473 nm light stimulation for 5 s periods (10–20 mW, 20 Hz, 5 ms pulses; 65 s interstimulation interval). Multiunit activity (MUA) was recorded during the 5 s of stimulation and was compared with the 5 s that preceded and followed the laser stimulation. (D) Example MUA at recording sites expressing ChETA-eYFP. (E) Average firing rate over multiple recording sites in the NAc expressing ChETA-eYFP (green, n = 20; 3 rats) or eYFP (blue, n = 16; 3 rats). Firing rate was normalized to pre-stimulation activity for each recording site (main effect of group, p<0.0001). 'n' represents the number of recording sites sampled. Statistical significance was calculated with Kruskal-Wallis followed by paired planned comparisons. All data are mean ± s. e.m. ***p<0.001.

to apply the laser stimulation during the last two of the four extinction sessions because this late stage corresponded to the time points at which we found that reward counterconditioning reduced the subsequent return of fear (*Figure 2D*, *Figure 2—figure supplement 1B–C* and *Figure 2—figure supplement 2E–G*). Finally, we tested fear memory recall 60 days after fear conditioning in order to determine whether enhancing activity in the BLA-NAc circuit during fear extinction acquisition could alter the persistence of extinction memory over weeks.

The ChETA-eYFP and eYFP-only groups acquired equivalent levels of fear conditioning and extinction learning (*Figure 5C*; Days 1–5). We observed no immediate effects of the laser treatment

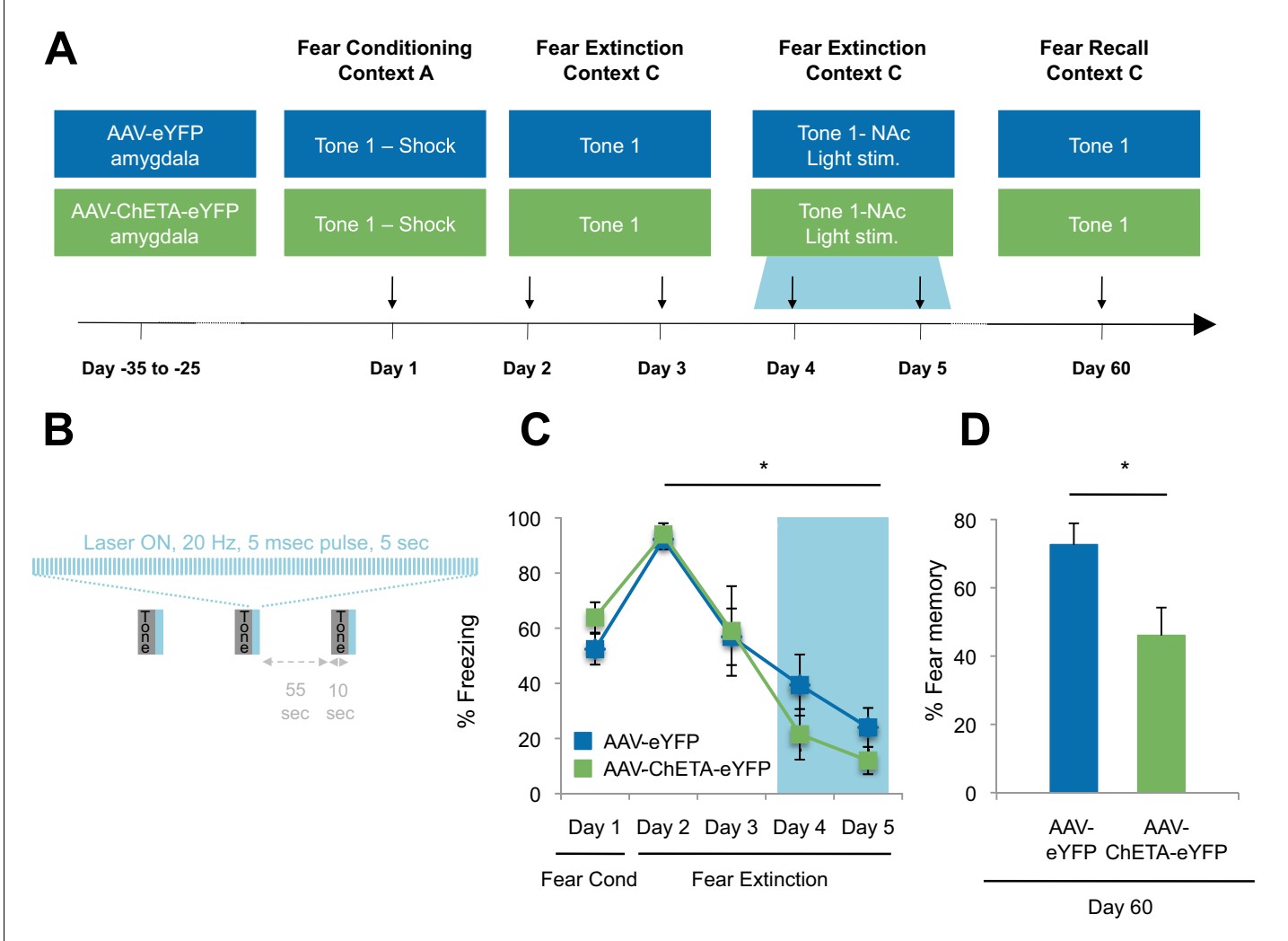

**Figure 5.** Optogenetic stimulation of the amygdala-NAc circuit during extinction of fear impairs the return of fear. (**A**) Experimental design (eYFP, n = 9; ChETA-eYFP, n = 4). (**B**) The timing of each pulse of light delivery is depicted. (**C**) Fear to the tone was measured as the percent of time spent freezing during the first five tone presentation trials of each behavioral session (main effect of group, p=*n.s.*; for Days 2 to 5: main effect of day, p<0.05). (**D**) Fear return on Day 60 was calculated as the percent of time spent freezing on the first five tone presentation trials on Day 60 normalized (per rat) to the percent of time spent freezing on the first five tone presentations on Day 2 (p<0.05). 'n' represents the number of animals. Statistical significance was determined by two-tail t-test for unpaired data (**C**) and two-tail Mann-Whitney test (**B**). All data are mean ± s.e.m. *p<0.05.

The following figure supplements are available for figure 5:

**Figure supplement 1.** Optogenetic stimulation of amygdala-NAc circuitry can promote reward seeking.

**Figure supplement 2.** Optogenetic stimulation of amygdala-NAc circuitry during extinction of fear impairs later fear recall.

during the two sessions of training when light stimulation followed tone presentation (*Figure 5—figure supplement 2A–B*). Strikingly, when we tested the rats again, without further experimental intervention 60 days after fear conditioning, the rats that had received optogenetic stimulation of the ChETA-eYFP-expressing BLA terminals in the NAc exhibited significantly reduced freezing behavior relative to control rats expressing eYFP alone (*Figure 5D* and *Figure 5—figure supplement 2C*). These results demonstrate that activation of the BLA-NAc circuit during fear extinction can significantly reduce the return of fear at a remote time point without affecting the acquisition of fear

extinction, thus providing causal evidence that activity in the specialized BLA-NAc circuit promotes the persistence of fear extinction memory.

## Activation of a BLA-NAc circuit increases activity in the infralimbic cortex but not in the BLA

The prelimbic (PL) and infralimbic (IL) areas of the medial prefrontal cortex (mPFC) are known to modulate fear, with the PL promoting fear learning and the IL promoting fear extinction (*Sotres-Bayon and Quirk, 2010*; *Senn et al., 2014*). If BLA-induced activation of the NAc led to circuit-level effects, these cortical regions might themselves be affected by terminal stimulation of BLA-NAc afferents. To test for such potential effects, we used cFos immunolabeling to estimate potential activation of these cortical regions (*Figure 6A*). Because the projections from the BLA to the NAc are predominantly ipsilateral (*Christie et al., 1985*), we stimulated the BLA terminals in the NAc of one hemisphere in anesthetized rats [10–20 mW, 473 nm, 20 Hz, 5 ms pulses, 65 s interstimulation interval, matching the parameters of the prior experiment in which optogenetic stimulation was administered during fear extinction (*Figure 5*)] (*Figure 6—figure supplement 1*) but visualized immunolabeling bilaterally to compare the effects of stimulation on the ipsilateral and contralateral mPFC. The IL ipsilateral and contralateral to the stimulation site had significantly higher numbers of cFos+ cells than found in naïve rats (*Figure 6C*). This cFos activation was especially pronounced in the ipsilateral IL (*Figure 6C*). In the PL, the number of cFos+ cells was significantly greater ipsilateral, but not contralateral, to the side of stimulation, relative to the naïve controls, (*Figure 6D*) though the overall magnitude of activation was much lower than that observed in the IL. These results suggested that BLA-NAc stimulation can result in activation of the medial wall cortex, especially the IL, and, to a lesser extent, the adjoining PL.

To determine whether activity in the NAc is necessary for the BLA-NAc stimulation-induced activation of the IL, we temporarily inactivated the NAc prior to optogenetic stimulation. Anesthetized rats received bilateral intra-NAc infusion of muscimol-bodipy (*Figure 6B*) 20–40 min before bilateral stimulation of the BLA-NAc (10–20 mW, 473 nm, 20 Hz, 5 ms pulses, 65 s interstimulation interval). Muscimol infusion significantly reduced the number of cFos+ cells in the IL, as compared to the number observed following stimulation in the absence of muscimol (*Figure 6C*). Altogether, these results indicated that BLA-NAc stimulation results in activation of the IL, which is mediated, at least in part, by activity within the NAc.

Given our evidence that the BLA-NAc circuit is especially activated when reward conditioning was paired with extinction training, we next asked whether reward conditioning during fear extinction could also result in increased activity in the IL. We performed cFos immunolabeling of sections containing the mPFC from rats in the *Ext-Ext* and *Ext-RC* groups from *Figure 3A*. We found significantly higher numbers of cFos+ neurons in the IL in the *Ext-RC* group than in the *Ext-Ext* group (*Figure 6E*), and we did not observe a difference in the PL (*Figure 6F*).

These results suggest that activation of the BLA-NAc circuit results in activation of the IL, a brain region widely implicated in fear extinction learning (*Sotres-Bayon and Quirk, 2010*), and that this BLA-NAc circuit activation affected the PL, implicated in fear learning, to a lesser degree. These findings raise the possibility that activation of the NAc during fear extinction may facilitate the recruitment of a cortical brain circuit that promotes the decrease of long-term fear expression.

Targeting terminals of BLA neurons in the NAc could also lead to antidromic activation of the BLA itself. We tested for this possibility by performing *in vivo* recordings and cFos immunostaining in the BLA of anesthetized rats in which we optogenetically stimulated BLA terminals in the NAc expressing ChETA-eYFP (*Figure 6—figure supplement 2A*). This stimulation did not alter multiunit activity in the BLA ipsilateral to the stimulation site in the NAc (*Figure 6—figure supplement 2B*). Since the projections from the BLA to the NAc are predominantly ipsilateral (*Christie et al., 1985*), we stimulated the terminals from the BLA to the NAc unilaterally and imaged bilaterally to compare the effects of stimulation in the ipsilateral and contralateral BLA. The number of cFos+ cells in the BLA ipsilateral to the BLA-NAc optogenetic stimulation site was not significantly different from the BLA in naïve rats (*Figure 6—figure supplement 2C*), nor was it significantly different in the BLA contralateral to the stimulation site (*Figure 6G*). This result indicates that the stimulation we used does not significantly activate BLA neurons. However, it is possible that this stimulation results in selective antidromic activation of BLA neurons that express ChETA-eYFP.

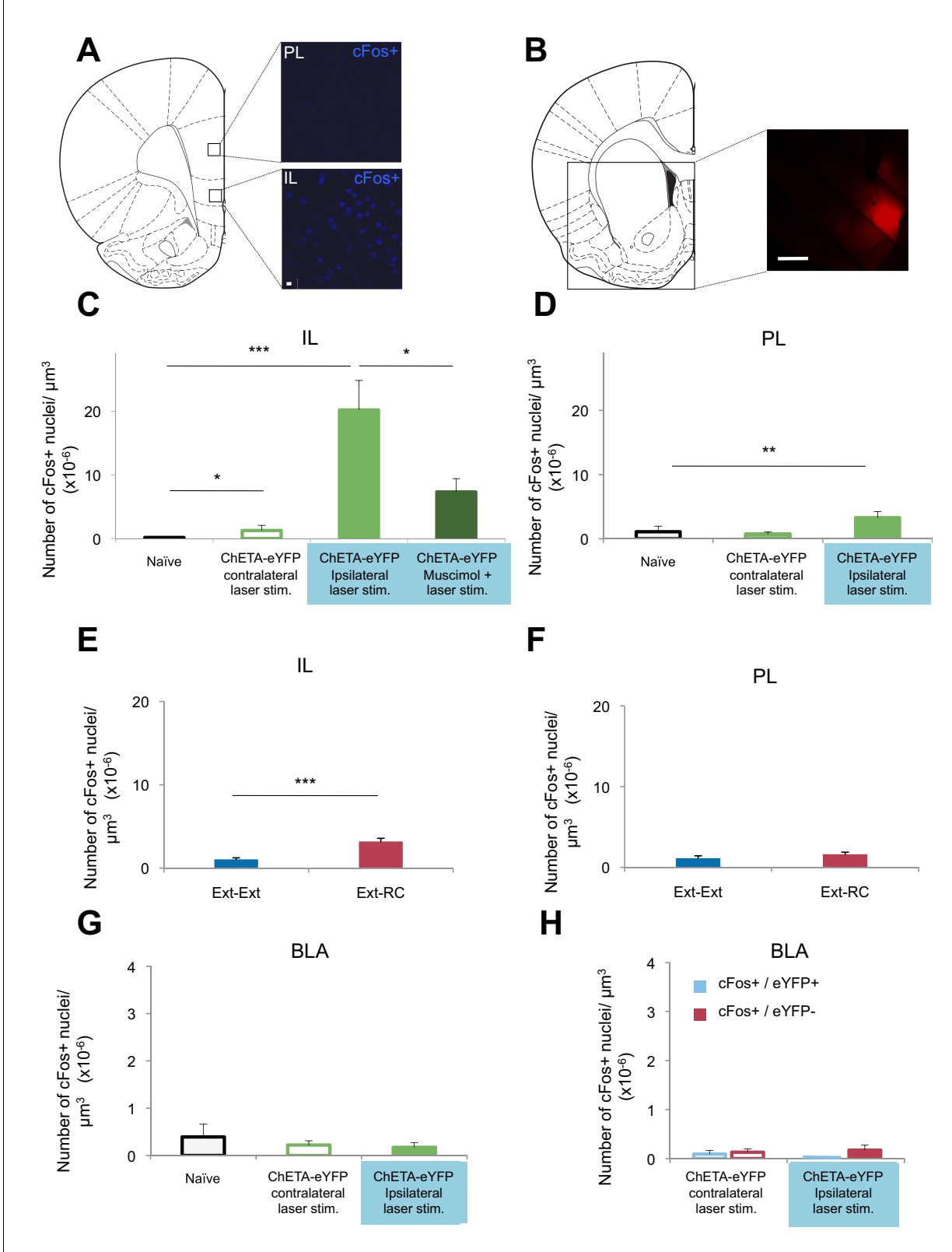

**Figure 6.** Optogenetic stimulation of an amygdala-NAc circuit or reward conditioning during fear extinction increases activity in IL. (A–D,G,H) Rats were infused with AAV-ChETA-eYFP in the amygdala. Anesthetized rats were subjected to 25 trials of stimulation of amygdala terminals within the NAc with 473 nm laser light for five-second periods, per trial (10–20 mW, 20 Hz, 5 ms pulses; 65 s interstimulation interval). (A, C, D) Coronal slices encompassing the infralimbic (IL) and prelimbic (PL) cortex were stained for cFos protein. (A) Representative image of cFos staining in the IL and PL. (B) Some rats also

*Figure 6 continued on next page*

*Figure 6 continued*

received infusion of muscimol-bodipy into the NAc. Representative image of muscimol-bodipy infusion in the NAc. (**C, D**) Numbers of cFos+ cells (*Naïve*, C: n = 14, D: n = 18, from the same 3 rats presented in *Figure 1C*; AAV-ChETA-eYFP contralateral to the stimulation site, C: n = 9, D: n = 10 from 5 rats; AAV-ChETA-eYFP ipsilateral to the stimulation site, C: n = 10, D: n = 10, from 5–6 rats; AAV-ChETA-eYFP + Muscimol, ipsilateral to the stimulation site, C: n = 15 from 3 rats) were counted and plotted per $\mu m^3$. (**E, F**) Coronal slices encompassing the IL and PL from animals trained as indicated in *Figure 3A*, were stained for cFos protein and numbers of cFos+ cells were counted and plotted per $\mu m^3$ (*Ext-RC* group, E: n = 14 from 5 rats, F: n = 12 from 5 rats; *Ext-Ext*, E: n = 11 from 3 rats, F: n = 9 from 3 rats). (**G, H**) Coronal slices encompassing the amygdala were stained for cFos protein (*Naïve*, n = 34 different slices from the same 3 rats presented in *Figure 1C*; AAV-ChETA-eYFP contralateral to the stimulation site, n = 18 from 4 rats; AAV-ChETA-eYFP ipsilateral to the stimulation site, n = 21 from 4 rats). Numbers of cFos+ cells (**G**), cFos+ and eYFP+ (**H**), and cFos+ and eYFP− cells (**H**) were counted in the BLA ipsilateral (n = 21 from 4 rats) and contralateral to the NAc stimulation site (H: n = 18 from 4 rats) and plotted per $\mu m^3$, in the BLA. 'n' represents the number of images analyzed (**C–H**). Scale bar represents 10 $\mu m$ (**A**) or 1 mm (**B**). Statistical significance was calculated with two-tail Mann-Whitney test (**E, F, H**), Kruskal-Wallis test followed by the Mann-Whitney test (**C, F, G**). All data are mean ± s.e.m.

The following figure supplements are available for figure 6:

**Figure supplement 1.** Location of fiber optic tips in rats receiving BLA-NAc optogenetic stimulation while anesthetized.

**Figure supplement 2.** Optogenetic stimulation of BLA presynaptic terminals projecting to NAc in anesthetized rats.

To test this hypothesis, we used a double-labeling strategy to search for BLA neurons that were active (cFos+) and that express ChETA-eYFP (eYFP+). The number of cells double-positive for cFos and eYFP in the BLA was similarly low in the BLA ipsilateral to the stimulation site as compared to the BLA contralateral to the stimulation site (*Figure 6H*). Also, we did not observe selective activation of BLA cells that express ChETA-eYFP (eYFP+): the number of cFos+ and eYFP+ double-positive cells was not higher than the cells that were both cFos+ and eYFP- in the BLA (*Figure 6H*).

In summary, we found no change in the firing rates of BLA neurons, nor changes of cFos labeling in the BLA during optogenetic stimulation of a BLA-NAc circuit. These results indicate that we likely achieved activation of the BLA-NAc projection terminals without major alterations of spike activity in the BLA.

## Discussion

We found that a BLA-NAc pathway is activated by fear conditioning and fear extinction training. The enhanced recruitment of this circuit during extinction learning, achieved either by pairing reward with fear extinction training or by optogenetic stimulation of this circuit during fear extinction, increases the activation of the IL and reduces the return of fear months after fear extinction.

While activation of this pathway by extinction training scales proportionally with activation of the BLA by extinction training (*Figure 1*), additional findings also bolster the claim that extinction training recruits the BLA-NAc circuit. The recruitment of this pathway by extinction is strongly related to the degree of extinction (*Figure 1—figure supplement 2*) and activation of the BLA-NAc projection deepens long-term extinction memory (*Figure 5*). If activation of the BLA-NAc pathway was simply the result of chance, then one would not expect to observe a relationship between the magnitude of extinction learning and activation within this pathway.

This evidence provides support for the proposal that fear extinction may engage an opponent-process reward system (*Luo et al., 2015*), as suggested by studies in human subjects in which ventral striatum activations were observed on trials on which the aversive stimulus (shock) was omitted in fear extinction sessions (*Raczka et al., 2011*). Exposure therapy combined with recall of pleasant memories has been successfully used to treat PTSD (*Hawk and Abel, 2011*), but the mechanism by which this method achieves therapeutic efficacy and the relative efficacy of this treatment as compared to exposure therapy are not known. Previous rodent studies have tested the effect of reward conditioning on avoidance behavior shortly after fear extinction (24 hr), but these studies have yielded conflicting results (*Nelson, 1966*; *Richardson et al., 1982*) and did not address long-term fear recall at remote time points. We found here that long-term spontaneous recovery of fear was significantly reduced when rats received reward conditioning combined with fear extinction, as compared to rats that received only extinction training following fear conditioning.

It may seem counterintuitive that the BLA-NAc pathway is similarly activated by the opposing behaviors induced by fear conditioning and fear extinction (*Figure 1—figure supplement 2D*). One possibility is that fear conditioning and fear extinction activate valence-segregated populations of NAc-projecting cells within the BLA (*Correia and Goosens, 2016*). However, our observations that indiscriminate optogenetic stimulation of BLA terminals in the NAc promotes reward-seeking (*Figure 5—figure supplement 1D*) and the weakening of fear memory (*Figure 5D*) suggests that if such valence segregation exists, appetitive subcircuits dominate, consistent with previous studies (*Namburi et al., 2015*). An alternative possibility is suggested by the observation that cFos expression is induced by both synaptic strengthening and synaptic weakening (*Lindecke et al., 2006*; *Kemp et al., 2013*). By this logic, cFos-related activation of the BLA-NAc pathway by fear conditioning may indicate synaptic weakening within the BLA-NAc circuit. Although activation of the BLA-NAc pathway is not selective for fear extinction, our data nonetheless support an important and novel role for this pathway in the long-term reduction of fear memory.

Neither reward conditioning during fear extinction nor optogenetic activation of the BLA-NAc circuit during fear extinction led to changes in across-session acquisition of fear extinction on Days 4 and 5, although it is possible that the overall low levels of freezing observed on Days 4 and 5 could make such a difference difficult to detect. However, both manipulations result in a significant reduction of fear expression two months after fear conditioning. Little is known about how fear recovers across time after extinction; future studies aimed at understanding this complex process and the BLA-NAc circuit effects on limbic networks could help to understand the efficacy of fear extinction-based treatments in humans.

The magnitude of the effects reported here are similar to those observed following manipulations of the IL region highlighted by our findings and known to regulate fear extinction memory (*Sotres-Bayon and Quirk, 2010*). The decrease in fear recall observed in rats that experience fear extinction with reward conditioning is comparable to the decrease of fear recall in animals subjected to microstimulation of the IL during fear extinction: in previous studies, IL activity was positively correlated with fear extinction recall (*Milad and Quirk, 2002*). We found here that optogenetic stimulation of the BLA-NAc circuit and also reward conditioning in combination with fear extinction lead to increases in activity, measured by cFos expression, in IL as well as in NAc itself, suggesting that these two brain regions may cooperate to decrease fear expression through mechanisms yet to be fully understood.

We did not find the optogenetic stimulation of BLA terminals in the NAc to be inherently rewarding (*Figure 5—figure supplement 1A*), as shown in previous experiments in mice (*Britt et al., 2012*; *Stuber et al., 2011*). One possibility is that the optogenetic stimulation we applied targeted a smaller or specialized part of the NAc: the estimated volume of NAc in the rat is more than double that of the mouse, and the NAc is a heterogeneous structure. We did find, however, that stimulation of the BLA-NAc projection can increase reward-seeking (*Figure 5—figure supplement 1C–D*), which is consistent with the idea that the BLA-NAc circuit exerts a reward-related modulatory effect upon emotional learning.

Our findings demonstrate that the administration of reward conditioning during fear extinction training also reduces the return of fear memory approximately two months after fear conditioning. We show that such enhancement of the enduring reduction of fear could be induced by the engagement of the BLA-NAc circuit during fear extinction training. Connections from the BLA to the NAc have already been recognized as important for generating goal-directed behavior in response to appetitive cues (*Ambroggi et al., 2008*; *Ghitza et al., 2003*; *Wan and Peoples, 2006*; *Day et al., 2006*; *Nicola et al., 2004a*, *2004b*). For example, by using a cue discrimination task similar to the one used in our reward conditioning paradigm, it has been shown that the BLA drives NAc neurons to encode the value of a reward-predictive cue (*Ambroggi et al., 2008*). It is possible that such cue-related plasticity is involved in the BLA-NAc circuit effects that we demonstrate here. Testing this idea by pharmacological inhibition of the BLA or temporally specific inhibition of the BLA-NAc circuit using cue-paired optogenetic silencing during fear extinction with reward conditioning is a natural further step, but these manipulations would produce ambiguous results because they would interfere with the successful acquisition of the rewarding conditioning by reducing or eliminating responding to the reward predictive cue (*Ambroggi et al., 2008*; *Stuber et al., 2011*). Thus, determining the cellular and molecular mechanisms underlying the contribution of the BLA-NAc circuit activity during fear extinction with reward conditioning to the persistent reduction of fear remains to be determined.

We did not determine whether different populations of neurons are recruited by fear extinction with reward conditioning versus reward conditioning alone. An interesting possibility raised by our findings is that fear extinction and fear extinction combined with reward conditioning activate substantially overlapping populations of cells in the NAc, at least if the predictive cue is shared between the two experiences. In our experiments, fear extinction with reward conditioning was only effective when the tone paired with sucrose reward was the same as that previously paired with footshock (*Figure 2*,*Figure 2—figure supplements 1* and *Figure 2—figure supplement 2* versus *Figure 2—figure supplement 3*). Even lacking a full understanding of NAc single neuron encoding of aversion, reward, or the absence of expected reinforcement, our findings suggest that fear extinction in which no explicit reward was present activates a BLA-NAc circuit, as does reward conditioning, but less effectively.

The NAc, the target of the BLA-NAc circuit identified here, regulates both aversive and appetitive behaviors, and evidence suggests that the NAc regions modulating these opposing behaviors are partially segregated along a rostro-caudal axis (*Basso and Kelley, 1999*). The rostral-caudal border separating reward-related and aversive-related processing within the NAc is dynamic and altered by experience; for instance, stressful environments cause caudal fear-generating zones to expand rostrally (*Reynolds and Berridge, 2008*). Because emotional experience changes across the fear extinction training, it is difficult to assess systematically whether recruitment of a BLA-NAc circuit by extinction varies along the rostral-caudal axis of the NAc, but our findings do indicate that the NAc can exert a modulatory role during fear extinction through activity of a BLA-NAc circuit. Of note, high frequency stimulation of the BLA produces synaptic depression in rostral NAc, but synaptic potentiation in caudal NAc (*Gill and Grace, 2011*), and these synaptic changes are thought, respectively, to reduce fearful behaviors and to promote appetitive states (*Reynolds and Berridge, 2008*). Because the presynaptic terminals originating in the BLA are broadly distributed across the NAc, it is possible that the BLA-NAc circuit biases the appetitive-aversive balance towards an appetitive state (*Britt et al., 2012*; *Stuber et al., 2011*). Our observation that BLA-NAc optogenetic stimulation promotes reward-seeking (*Figure 5—figure supplement 1C–D*) is consistent with this interpretation.

The reduction of long-term fear memory that we found by adding reward conditioning to extinction training could be due to increased exploratory behavior or elevated blood glucose levels, but our findings do not favor these possibilities. Optogenetic stimulation of the BLA-NAc pathway produced a reduction of long-term fear memory similar in size to the reduction of long-term fear memory that we found by adding reward conditioning to extinction training. These results suggest that reward conditioning during fear extinction did not enhance extinction learning simply by promoting exploration, as there was no reward port in the optogenetic experiment to induce behaviors that could compete with freezing responses. Our optogenetic findings also suggest that reward conditioning during fear extinction did not enhance extinction learning by increasing blood glucose levels (*Gold et al., 2012*). Rats receiving optogenetic stimulation did not receive reward (sucrose), and rats that did receive reward (sucrose) conditioning with a novel tone never paired with shock did not exhibit decreased fear recall. Our results instead favor the novel view that activation of the BLA-NAc circuit during extinction of fear reduces fear recall by strengthening extinction memory *per se*.

Potential confounds of the projection optogenetics strategy that we used include back-propagation of ChETA-induced action potentials or activation of fibers of passage running through the NAc region targeted by the laser illumination. We used optogenetic stimulation of the BLA-NAc circuit in anesthetized rats to examine the effect of this stimulation on different brain areas in the absence of behavioral confounds (*Figure 6*; *Figure 6—figure supplement 2*). Our results suggest that back-propagation was unlikely to be responsible for the effects of stimulation on behavior because the laser stimulation in the NAc did not alter multiunit activity in the BLA ipsilateral to the stimulation site or the number of cFos+ nuclei in the BLA. Also, muscimol infusion in the NAc reduced IL activation mediated by BLA-NAc stimulation. These results indicate that optogenetic stimulation-induced activity in the NAc, contributes, at least in part, to the increased activity in the IL, a region known to modulate fear extinction. These results in anesthetized animals parallel our findings in awake, behaving rats: reward conditioning combined with fear extinction produced activation of the IL and, to a lesser extent, the PL (*Figure 6*). In addition, other studies have shown that stimulation of the NAc in awake animals activates the IL (*Vassoler et al., 2013*), as we report here in anesthetized rats. However, the level of neuronal activation measured in the anesthetized rats does not reflect the level of neuronal activation in awake, behaving rats.

While anesthesia eliminates variability in cFos expression attributable to behavior, it suffers from a different set of caveats. Anesthesia alters the balance of neuronal excitation and inhibition (*Hentschke et al., 2005*), which could result in either false positive cFos signal (due to disinhibition) or a reduction in cFos signal (due to anesthesia-induced inhibition). Our data (*Figure 6G*) show that anesthetized and naïve, awake rats express similar levels of cFos protein in the BLA, suggesting that it is unlikely that isoflurane anesthesia induces false positive cFos signal in the BLA. Nonetheless, it is possible that anesthesia could increase activity of inhibitory interneurons in the BLA and therefore mask potential antidromic activation of BLA neurons. Therefore, we must acknowledge that in the experiments where optogenetic stimulation was delivered to anesthetized rats, the level of neuronal activation reported in the BLA, NAc, IL and PL is likely to be different from what would be measured in awake animals.

In addition, we cannot rule out a contribution by fibers of passage or collateral fibers stimulated by laser treatment of the BLA-NAc circuit. The ventral striatum contains fiber bundles from the ventral mPFC (a homolog of IL) and the dorsal anterior cingulate cortex (a homolog to PL) to the amygdala and thalamus (*Lehman et al., 2011*). Additionally, some BLA neurons projecting to the NAc have axon collaterals within the mPFC (*McDonald, 1991b*; *Shinonaga et al., 1994*). Therefore, it is possible that activation of these collateral projections or fibers of passage could partially account for the increased activation of the IL as a result of optogenetic stimulation of the BLA-NAc circuit or of reward conditioning in combination with fear extinction. However our results clearly show that NAc activation itself contributes to the activation of the IL following stimulation of the BLA-NAc circuit.

An additional problem of our optogenetic strategy is that we undoubtedly activated a greater proportion of the BLA-NAc pathway with stimulation than is activated by fear conditioning or fear extinction. This is supported by the observation that the NAc shows greater expression of Nr4a3 after stimulation than is observed after extinction training alone (*Figure 3—figure supplement 1F*). However, the degree to which our stimulation produced greater activation of this pathway than natural extinction training is difficult to quantify for several reasons, including: 1) Viral expression of the opsin targeted only a portion of the BLA, and light delivery targeted only a portion of the NAc, thus only a fraction of the BLA terminals within the NAc could be stimulated, and 2) Our CTB infusions targeted only a small portion of the NAc, and thus identified only a subset of the BLA cells that project to this area. By using activity-dependent expression of channelrhodopsin, as has been used in other experimental contexts (*Redondo et al., 2014*; *Gore et al., 2015*), we could more precisely enhance activation of the extinction-related portion of the BLA-NAc circuit. In the absence of such a strategy, optogenetic stimulation may induce supra- or non-physiological changes. For example, the degree of activation that we observed in IL after optogenetic stimulation (*Figure 6C*) might be much greater than is achieved by any behavior. Also, our non-specific strategy may not only have activated BLA-NAc circuits that underlie fear extinction, but also those that may be activated by fear conditioning (*Figure 1—figure supplement 2*). Nevertheless, the fact that optogenetic stimulation produces a similar enhancement to the persistence of fear extinction memory (*Figure 5D*) as reward conditioning (*Figure 2D*) supports our claim that reward conditioning deepens the persistence of fear extinction memory via more effective recruitment of the BLA-NAc circuit than extinction alone.

Some of our experiments characterizing the impact of BLA terminal stimulation in the NAc were conducted in animals anesthetized with isoflurane (*Figures 4*, *6*). While some studies have shown that isoflurane anesthesia has no impact on reward-related induction of cFos in the NAc (*Kufahl et al., 2015*) and that it does not induce cFos expression in either the NAc or IL (*Smith et al., 2016*), many studies clearly show that anesthesia does alter neuronal properties (*Chen et al., 2011*; *Purtell et al., 2015*; *Joksovic and Todorovic, 2010*; *Ogawa et al., 1992*). Therefore, we must acknowledge the possibility that the magnitude of the stimulation-induced changes that we report are different from what occurs in an awake, behaving animal.

Deep brain stimulation (DBS) in the NAc region has been found to reduce fear recall (*Whittle et al., 2013*; *Rodriguez-Romaguera et al., 2012*). It is still unclear if DBS therapeutic effects are due to increases or decreases in neuronal activity (*Nauczyciel et al., 2013*). Using optogenetics, we found that stimulating BLA terminals within the NAc increased spike activity in the NAc (*Figures 4D–E*). Such activation accords with other optogenetic studies of BLA projection activation of BLA-NAc connections (*Britt et al., 2012*; *Stuber et al., 2011*), in which the optical stimulation of the BLA-NAc circuit results in local glutamate receptor-mediated excitatory postsynaptic currents in NAc neurons. Since local glutamate antagonism within the NAc elicits appetitive or aversive

behaviors on its own (*Reynolds and Berridge, 2008*), the role of NAc glutamate receptors during optogenetic stimulation of the BLA-NAc circuit is difficult to assess.

Our findings raise the possibility that recruitment of a selective BLA-NAc circuit contributes to the mechanism by which counterconditioning achieves therapeutic efficacy. It may also explain a recent finding that replacing expected threat with a neutral, unexpected outcome also enhanced the short-term persistence of extinction (*Dunsmoor et al., 2015*). These results raise the possibility that the BLA-NAc circuit could be a valuable target for therapeutic intervention across a range of disorders in which extinction-based therapies are applied.

## Materials and methods

### Subjects

All experiments used adult male Long-Evans rats (250–350 g, Taconic, Germantown, NY or Harlan, Indianapolis, IN), housed individually (20–22.2°C; 12-hr light-dark cycle, lights on from 6:45 am to 6:45 pm). Food and water was provided *ad libitum* until 11 days before the start of the behavioral experiment; subsequently, food was limited to 4 g per 100 g of body weight daily for each rat. All procedures were in accordance with the US National Institutes of Health (NIH) Guide for the Care and Use of Laboratory Animals and were approved by the MIT Institutional Animal Care and Use Committee (protocol 0313-018-16) and the Animal Care and Use Review Office of the USAMRMC (proposal 58076-LS-DRP.01).

### Viral infusions and fiber optic implants

Rats were anesthetized with 1.5% isoflurane vaporized in oxygen and mounted into a dual arm stereotaxic frame (Kopf Instruments). AAV-eYFP or AAV-ChETA-eYFP viruses (both at 1.3 x 10¹³ vg/ml) were bilaterally infused (1 µl/side) in the amygdala (A/P −2.3, M/L ± 4.9, D/V −7.3) using 31-gauge needles attached to Hamilton syringes. The infusion was carried out at a rate of 0.1 µl/min, and the needle was slowly removed 20 min after the end of infusion. Bilateral fiber optic implants aimed at the nucleus accumbens (A/P +1.8, M/L ± 1.2, D/V −5.7) were secured by the placement of three jeweler screws in the skull and dental acrylic. Rats were allowed to recover for one month after surgery before being subjected to auditory fear conditioning and extinction experiments.

### CTB-Alexa fluor 488 infusions

Rats were anesthetized with 1.5% isoflurane vaporized in oxygen and mounted into a dual arm stereotaxic frame (Kopf Instruments). CTB-Alexa 488 retrograde tracer (1 µl/side, Invitrogen) was bilaterally infused in the nucleus accumbens (A/P +1.8, M/L ± 1.2, D/V −6.4) using 31-gauge needles attached to Hamilton syringes. The infusion was carried out at a rate of 0.1 µl/min, and the needle was slowly removed 20 min after the end of infusion. Rats were allowed to recover for 11 days after surgery before being subjected to auditory fear conditioning and extinction experiments.

### Pavlovian fear conditioning

Fear conditioning experiments were conducted in a modified chamber (Med Associates) housed in a sound-attenuated cubicle. The animals were placed in individual chambers and video of each session was recorded. Each experiment used auditory fear conditioning wherein rats received 5 pairings of tone (2 KHz, 8 KHz or white noise, 85dB, 10 s) and footshock (1 s, 0.6 mA) in a unique context (context A: metal shock grid floors, chamber fan on, 1% acetic acid odor, house and room lights off). Animals were allowed 3 min to habituate to the chamber before tone-footshock pairings (5 pairings total) were given at intervals of 1 min (tone exposure groups were subjected to the same context and tone but no footshock was administered). Thus, each fear conditioning session lasted a total of 8 min 50 s.

Fear extinction was conducted 1, 2, 3, 4 and 55 (or 60) days after fear conditioning by placing the animals in a novel context (context B: white Plexiglas plastic floors, curved Plexiglas wall inserts, fans off, 0.3% Pine-Sol odor, house and room lights on or context C: white Plexiglas box, 0.3% Pine-Sol odor and room lights on). Two minutes after placement in context B or C, fear to the tone was assessed by presenting multiple discrete tones (10 for plasma corticosterone measurement experiments and 25 for all other experiments), with a 10 s duration and 1 min inter-stimulus interval. In some cases, extinction sessions consisted of presentation of two different tones (as indicated in the

figures; one of the tones was the same as used during the fear conditioning session and the other was a second novel tone). During the fear extinction sessions with two tones, the two tones were presented in a pseudorandom order (25 presentations of each tone). For tone or context exposure sessions, rats were subjected to the same tone and context or context only, respectively. Thus, for extinction sessions in which only one tone was presented, the total length of the session was 31 min, 16 s. For extinction sessions in which two tones were presented, the total length of the session was 73 min, 30 s. For extinction sessions in which plasma corticosterone was measured, the total length of the session was 13 min, 10 s.

For some experiments, reward conditioning was used during the fear extinction sessions in Context B (*Figure 2*, *Figure 2—figure supplement 2*, *Figure 3*). In these experiments, the reward port was available to all animals in all groups during all sessions conducted in Context B, however sucrose was delivered through this port only to a subset of these groups in a subset of these sessions (as indicated in the respective experimental design panels). The reward port was never available during the fear conditioning in Context A.

Freezing was measured using commercial software (VideoFreeze, Med Associates), automated video analysis (MATLAB) or human scoring (performed by people blind to the experimental conditions). Raw freezing levels, computed as the percent of the relevant time period spent freezing, is depicted in panels labeled '% Freezing'. The '% fear memory' measure was calculated to assess the spontaneous recovery of fear at remote time points (Day 55 or Day 60) following extinction training. The% fear memory recovered on Day 55 or 60 was expressed as a percentage of the original fear memory acquired according to the following formula: (% freezing on Day 55 or 60 × 100) ÷ (% freezing of Day 2). The % freezing on Day 55 or 60 was calculated as % freezing during all Tone 1 trials on the first 5 trials on Day 55 or 60; the % freezing on Day 2 was calculated as the% freezing on the first 5 trials of Tone 1 presentations on Day 2 (thus reflecting the maximum fear memory acquired). Due to lack of equipment, video recordings (and thus freezing measures) on Days 4 and 5 were not captured for a subset of the animals whose data is illustrated in *Figure 2—figure supplement 2B*. Due to the variability in fear learning behavior for the dataset depicted in *Figure 2* and *Figure 2—figure supplement 1*, we excluded animals from both experimental groups (*Ext-Ext,* n = 16 rats, *Ext-RC* n = 6 rats with freezing levels below 80% during the first five trials with Tone 1 presentation of the fear extinction session on Day 2.

## Reward conditioning

Reward conditioning experiments were conducted in a modified chamber (Med Associates) housed in a sound-attenuated cubicle. The animals were placed in individual chambers and each experimental session used auditory reward conditioning wherein rats received 25 pairings of auditory stimuli (8 KHz or white noise, 85dB, 10 s) and reward (0.2 ml of 30% sucrose solution) and 25 presentations neutral auditory stimuli (8 KHz or white noise, 85dB, 10 s) in a unique context. The reward paired and neutral auditory stimuli (50 presentations in total) were presented in a pseudo-random order at intervals of 1 min. The total length of each session was 73 min, 30 s. Video was recorded throughout the session, and nose-pokes in the reward port were also recorded by the software.

Statistically significant decreases in the latency to approach the reward port, or increased reward port entries following the reward paired auditory stimuli as compared to the neutral stimuli, was used to identify rats that acquired reward conditioning. Rats that never displayed reward-seeking (failure to nose-poke) or discriminative reward conditioning (failure to nose-poke significantly more or faster to the CS paired with reward than to the neutral CS) were excluded from all analyses. This resulted in the following numbers for exclusion: *Figure 2* and *Figure 2—figure supplement 1*, n = 6. It is important to note that these excluded animals did not display different levels of fear either during conditioning or during the first two days of extinction training (data not shown). For data depicted in *Figure 2—figure supplement 1E*, we only included data from animals where the reward port was clearly visible on the video recordings and the experimenter was able to confidently score reward port entries.

Nose-poke latencies were computed by Med-PC software as the time elapsed between the onset of each tone and the first subsequent nose poke. Reward port entries were calculated by recording beam breaks at the reward port during the 10 s tones and during the 10 s that preceded each tone. Normalized reward port entries were computed according to the following formula: [average port entries during the 25 tones – average port entries during the 25 pre-tone periods (10 s before each

tone)]. Freezing behavior and% fear memory was measured as described above in *Pavlovian Fear Conditioning*.

## Optogenetic stimulation during fear extinction

Fear conditioning and extinction experiments were performed as described above. During the extinction training sessions on Days 4 and 5, tone. Tone presentations were paired with 473 nm light stimulation in the NAc for 5 s periods (10–20 mW, 20 Hz, 5 ms pulses; 65 s interstimulation interval).

## Optogenetic stimulation in anesthetized rats

Anesthetized rats expressing AAV-ChETA-eYFP in the amygdala (see 'Viral infusions and fiber optic implants' section for details) received 25 trials of 473 nm light stimulation in the NAc, either unilaterally or bilaterally. Each trial consisted of 5 s periods of light stimulation (10–20 mW, 20 Hz, 5 ms pulses; 65 s interstimulation interval) to match the previous experiment in which optogenetic stimulation was administered during fear extinction. A subset of rats received bilateral infusions of muscimol-bodipy (5 mM, 0.7 μl/side; Molecular Probes), in the NAc (A/P +1.8, M/L ± 1.2, D/V −6.4) using 35-gauge needles attached to Nanofil syringes, 20–40 min prior to the start of the optogenetic stimulation. One hour after the end of optogenetic stimulation, animals were anesthetized with an overdose of isoflurane and their brains were processed for immunohistochemistry (see *Immunohistochemistry* for details).

## Optogenetic stimulation during nose-poke-induced reward delivery

Rats were placed in an individual modified operant chambers (Med Associates) equipped with active and inactive nose-poke ports. Each active nose-poke performed by the animal resulted in delivery of 0.2 ml of a 30% sucrose solution and 5 ms pulses of 473 nm laser light delivered 100 times at 20 Hz. Each inactive nose-poke performed by the animal resulted in delivery of 0.2 ml of a 30% sucrose solution only. Both active and inactive nose-pokes triggered an audible tone (2 KHz, 2 s). Nose-pokes made within 3 s of an active nose-poke did not activate the laser. Active and inactive nose-poke timestamp data were recorded using MED-PC software.

## Optogenetic stimulation during place preference task

On Day 1, each rat was placed in the middle of a two chamber behavior box for 10 or 12 min. The two sides were differentiated by visual patterns on the walls. On Day 2, a divider was placed in the box to prevent rats from freely crossing between the chambers. Rats were placed in the least-preferred side of the box (determined by the amount of time spent on each side of the box on Day 1) while receiving BLA-NAc optogenetic stimulation (5 ms pulses, 20 Hz for 5 s every 20–30 s). The rats were then placed in the most-preferred side of the box for the same amount of time in the absence of stimulation. On Day 3, rats were again allowed to freely explore the two chamber behavioral box for the same amount of time as on Day 1. The place preference score was calculated as the difference between the percent of time spent (from total session time) in the side of the box paired with optogenetic stimulation on Day 3 and the percent of time spent in the same side on Day 1.

## Immunohistochemistry

Following completion of the experiments or 1h after the end of behavioral testing (for cFos and Nr4a3 staining in the BLA, medial NAc shell and mPFC) animals were anesthetized with an overdose of isoflurane and intracardially perfused with phosphate buffer solution (PBS, pH 7.4) followed by 4% paraformaldehyde solution in PBS. Brains were harvested and placed in 4% paraformaldehyde solution in PBS for 24 hr at 4°C. The brains were then transferred to a 30% sucrose solution for 3 days at 4°C. Coronal sections (40 μm) were mounted with Vectashield mounting media containing DAPI and imaged by either fluorescence or confocal microscopy or stored in cryoprotectant solution at −20°C until further processed for immunolabeling. Primary antibodies were rabbit anti-cFos (Calbiochem, PC38T for *Figure 1*, Santa Cruz sc-52 for *Figures 3*, *6* and *Figure 1—figure supplement 2*), and chicken anti-GFP (Invitrogen, A10262), rabbit anti-Nr4a3 (Santa Cruz, SC 30154) and mouse anti-synaptophysin (Sigma, S5768). Secondary antibodies were anti-rabbit conjugated to Alexa 594 or Alexa 405, anti-chicken conjugated to Alexa 488 and anti-mouse conjugated to Alexa 594. Images were obtained with a Zeiss confocal microscope (*Figures 1B* right panel, 3D and 6A were acquired with a

40x oil immersion lens, 0.7x zoom and Z-resolution of 2.5 µm; *Figure 4B* was acquired a 60x oil immersion lens, 4x zoom and Z-resolution of 0.2 µm; *Figure 6B* was acquired with 10x objective, 1x zoom and Z-resolution of 40 µm). Digital images were acquired using the LSM 510 software and were reconstructed and analyzed using Image J software. Other images were acquired with a Zeiss fluorescence with a 2.5x or 5x objective.

### *In vivo* electrophysiology

Rats infused with AAV-ChETA-eYFP or AAV-eYFP in the amygdala (as indicated above), were anesthetized with 1.5% isoflurane vaporized in oxygen and mounted into a dual arm stereotaxic frame (Kopf Instruments). Tungsten tetrodes were lowered into the BLA (A/P −2.3, M/L ± 4.9, D/V −6.5 to −7.4) while a fiber optic implant attached to tungsten tetrodes were lowered into the NAc (A/P +1.8, M/L ± 1.2, D/V −6 to −7.5). Electrical signals were amplified at 100–10000, sampled at 16 kHz, band-pass filtered for 600–6000 Hz, and recorded by a Cheetah data acquisition system (Neuralynx) while the optic fibers delivered 473 nm light stimulation for 5 s periods (10–20 mW, 20 Hz, 5 ms pulses; 65 s interstimulation interval). Firing rates preceding, during and following laser stimulation were calculated using MATLAB scripts.

### Corticosterone assay

Perimortem blood was collected from the trunk after decapitation in a tube which contained 1:100 v/v 0.5 M EDTA and 1:100 v/v HALT (Pierce). Immediately after collection, plasma was extracted by centrifugation (2100 g at 4°C for 10 min). The plasma layer was collected and stored at −80°C. Corticosterone levels were determined using a commercial ELISA (Enzo Life Sciences). Plasma was diluted 1:50 in assay buffer 15 and processed according to the manufacturer's protocol. Samples were excluded from analysis if they displayed signs of hemolysis or lipemia.

### RNA extraction and microarray processing

Total RNA was extracted from the NAc using Qiagen's miRNeasy Mini kit, and purified with on-column digestion of DNA using Qiagen's RNase-free DNase. Pooled samples were created by combining equal amounts of mRNA from six rats. The quantity and integrity of the extracted RNA was assessed using the Agilent 2100 Bioanalyzer, which confirmed that all RNA samples were of high quality (RNA Integrity Number ≥9.0). Gene expression profiling was accomplished by Affymetrix microarray Rat Gene 1.0 ST. All microarray processing was performed at the Whitehead Institute Genome Technology Core at MIT and analysis was done using Genespring software.

### Statistical methods

Statistical comparisons were calculated using Statview, MATLAB, or Excel. All data is expressed ± standard error of the mean. All group data were considered statistically significant if $p < 0.05$.

Subjects were randomly assigned to experimental groups prior to experimentation. No statistical methods were used to pre-determine sample sizes but our sample sizes are similar to those generally employed in the field. Animals with viral or tracer injections off target or fiber optic misplacement or damage during the experiment were excluded from the experiments. A small number of animals died from natural causes during the incubation time between the fear behavior on Days 1–5 and the remote fear test on Day 55 or 60. No data from any excluded animals was included in any analysis.

Data collection was not performed blind to the conditions of the experiments. However, imaging data analysis was largely performed using automated ImageJ routines; scoring of freezing behavior was mostly automated and in experiments where experimenters manually scored the data, the experimenters were unaware of animal group assignments.

## Acknowledgements

We thank Lindsay Johnson, Lindsay Kinney, Fangheng Zhou, Thomas Moulia, Norah Nguyen, Warren Slocum, Junmei Yao, Jordan Cruz, Cassandra Buzby, Connor Kirschbaum, Idil Ozturk, Dr. Ulf Knoblich, Dr. Geoff Schoenbaum and Dr. Donna Calu for technical support and Drs. Tyler Brown, Retsina Meyer, Kartik Ramamoorthi and Yasuo Kubota for discussions. We thank Dr. Li-Huei Tsai and Matt Dobbin for providing a confocal microscope and Dr. Alan Jasanoff for providing an ELISA plate

reader. This research was funded by NIMH (R01 MH084966), and the US Army Research Office and the Defense Advanced Research Projects Agency (grant W911NF-10-1-0059) to AMG and KAG.

## Additional information

### Funding

| Funder | Grant reference number | Author |
|--------|------------------------|--------|
| Defense Advanced Research Projects Agency | W911NF-10-1-0059 | Ann M Graybiel<br>Ki A Goosens |
| Army Research Office | | Ann M Graybiel<br>Ki A Goosens |
| National Institute of Mental Health | R01 MH084966 | Ki A Goosens |

The funders had no role in study design, data collection and interpretation, or the decision to submit the work for publication.

### Author contributions

SSC, Designed the experiments, Involved in all data collection and analysis, Interpreted data, Wrote the manuscript; AGM, Collected data and performed analysis of a subset of the imaging and behavioral experiments, Interpreted data; AL, Collected behavioral data and performed analysis of a subset of the imaging and behavioral experiments, Interpreted data; AMG, Interpreted data, Wrote the manuscript; KAG, Designed the experiments, Conducted behavioral analysis, Interpreted data, Wrote the manuscript

### Author ORCIDs

Ki A Goosens, http://orcid.org/0000-0002-5246-2261

### Ethics

Animal experimentation: All procedures were in accordance with the US National Institutes of Health (NIH) Guide for the Care and Use of Laboratory Animals and were approved by the MIT Institutional Animal Care and Use Committee (protocol 0313-018-16) and the Animal Care and Use Review Office of the USAMRMC (proposal 58076-LS-DRP.01).

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
