## [Decision Letter]

Thank you for submitting your work entitled "Amygdala-ventral striatum circuit activation decreases long-term fear" for consideration by *eLife*. Your article has been reviewed by three peer reviewers, and the evaluation has been overseen by Naoshige Uchida as the Reviewing Editor and a Senior Editor. C. Daniel Salzman (peer reviewer) has agreed to reveal his identity.

The reviewers have discussed the reviews with one another and the Reviewing Editor has drafted this decision to help you prepare a revised submission.

Summary:

This study examined the mechanism by which the extinction of fear memory could be facilitated. The authors show that counterconditioning, in which animals receive appetitive outcomes during fear extinction, enhances the retention of extinction memories, while having no effect on the initial extinction learning. They then show that the BLA-NAc circuit is recruited by extinction and that this recruitment is enhanced when extinction is done in combination with reward (counterconditioning). Furthermore, they demonstrate that direct stimulation of the BLA-NAc circuit enhances the retention of extinction memories.

Essential revisions:

All the reviewers felt that these results are very interesting and important. However, the reviewers also raised several substantive concerns. The main points to which authors need to address are summarized below:

1) Specificity of the recruitment of the BLA-NAc circuit during extinction and/or reward conditioning. It is not clear whether the increase in c-fos labeled BLA cells occurs preferentially in the NAc projecting cells or generally in the BLA. If this is occurring by chance, then the evidence for specific recruitment of the BLA-NAc pathway would not be provided. This point needs to be explicitly analyzed.

2) Data exclusions. It was noted that the authors excluded some samples/data, which might have biased some of the data analyses. The authors need to clarify this.

3) Potential non-physiological effects of optogenetic manipulation. The data suggests that 95% of the neurons activated by optogenetic manipulation were not activated by fear conditioning and extinction. The authors need to consider whether the effects they observe are non-physiological effects.

4) One of the reviewers raised various concerns in the experimental design and data analysis regarding the conclusion that the BLA-NAc circuit is recruited by fear extinction. See below for the details.

*Reviewer #1:*

This is an interesting paper on the role of a basolateral(BLA)-nucleus accumbens (NAc) circuit in fear extinction learning and counterconditioning in which animals undergo extinction in combination with appetitive learning. They show behaviorally that counterconditioning enhances the retention of extinction memories, while having no effect on the initial extinction learning. They then show that this pathway is recruited by extinction and that this recruitment is enhanced when extinction is done in combination with reward (counterconditioning). Furthermore, they demonstrate that direct stimulation of the BLA-NAc circuit enhances the retention of extinction memories. While the results are intriguing I have a number of substantive concerns which should be addressed:

1) For Figure 1, it is not clear whether the increase in cfos labeled BLA cells occurs preferentially in the NAc projecting cells or generally in the BLA. While the authors mention in the fifth paragraph of the subsection A BLA-NAc circuit is recruited by fear extinction that the a third of the cfos labeled cells are NAc projecting, they need to use a more rigorous statistical test to demonstrate that recruitment of this pathway is actually selective.

2) Related to the counterconditioning data presented in Figure 2, the authors mention in the Methods that "Rats that never displayed reward-seeking (failure to nose-poke) or discriminative reward conditioning (failure to nose-poke significantly more or faster to the CS paired with reward than to the neutral CS) were excluded from all analyses." This is potentially a major issue as they are selectively removing animals from one group. This could bias their results in the counterconditioning group such that the animals in this group have reduced fear specifically because these animals were removed and not because the counterconditioning enhanced extinction memory retention. For example, it is possible that animals that couldn't learn reward seeking had higher fear levels or more spontaneous recovery of extinction. The authors need to do something to address this problem. One possibility is to identify animals that would be excluded and then do extinction training and compare these animals that would have been excluded to a normal group of extinction animals to see if there are differences in spontaneous recovery. If not, then there is not a problem with the exclusion criteria.

Related to the above comment, they should state somewhere how many animals were excluded. Also, for the data in Figure 2, the authors should clarify whether this does or does not include excluded animals.

3) The authors should clarify in the Methods section whether the exact same box was used (including the availability of all apparatuses in box except reward) for both ext-ext and ext-RC groups? This is important as differences in the box could produce differences in extinction.

4) There are a number of issues with data presented in Figure 3:

A) It is not clear why there is an inset to compare Ext-RC and RC groups. Ext-Ext, Ext-RC and RC groups should be in a single graph with appropriate statistical comparisons.

B) A negative control group should be included here to determine whether there is an increase above baseline in fos activation. One of the control groups used in Figure 1 would be sufficient.

C) For the BLA data in 3B and C, representative images should be shown somewhere.

D) One possibility is that extinction (or reward) preferentially activates NAc projectors early in learning and that the differences you see between extinction and reward learning are due to that and not because the extinction and reward are summating to activate more cells. It would ideal to run another group where tissue is collected after 1 day of extinction to match the single day of reward learning.

E) Also need negative control for the NAc Nr4a3 data in Figure 3 as it is not clear whether there is a significant Nr4a3 activation above baseline in the ext-ext group.

5) For the data presented in Figure 5, they should show supplemental data with raw% freezing levels on day 2 and day 60 in supplement as they do for the data in Figure 2.

6) Though not absolutely necessary, inhibition of BLA-NAc terminals during extinction would be ideal to directly test whether the BLA-NAc projection participates causally in generating extinction. As it stands they present correlative evidence that the circuit is engaged, but the paper would much more compelling if they could make a causal link.

*Reviewer #2:*

This paper explores the role of the amygdala to ventral striatum (BLA to NAc) circuit in decreasing long-term fear. The paper makes a nice series of observations that emphasize the potential importance of this circuit -long recognized as mediating reward-related behavior in potentially modifying neural circuitry responsible for modulating fear responses on a long-term basis. Thus overall, I am supportive of publication of the paper. However, I have a number of issues that I think the authors should address to improve the paper.

1) Specificity of the recruitment of the BLA-NAc circuit during extinction and/or reward conditioning.

The authors' data largely rely on an observed increase following extinction or reward conditioning in double labeled cells in BLA using c-fos and CTB+ to mark activated cells that project to NAc. However, the authors do not address whether this increase in double labeling could be occurring by chance as a result of simply increasing the number of c-fos positive cells in BLA. In other words, increased double labeling would be expected simply by virtue of increasing the number of c-fos positive cells. If this is occurring by chance, then the evidence for specific recruitment of the BLA-NAc pathway would not be provided. Moreover, the statement that the proportion of activated BLA neurons that project to NAc increases with reward conditioning can't be supported without such statistical testing.

2) Potential non-physiological effects of optogenetic manipulation.

The authors report that only 5% of the NAc-projecting BLA neurons were double-labeled by c-fos and CTB+. This implies that 95% of the neurons activated by optogenetic manipulation were not activated by fear conditioning and extinction. The authors need to consider whether the effects they observe are non-physiological effects (essentially activating neurons not normally activated during the relevant behavior) given that they don't know much about the response properties of most of the neurons they are activating optogenetically. This type of issue applies to many (if not most) studies employing optogenetic methodology, but it is one that I think we all need to worry about more. A missing link in this paper is the specific activation only of those BLA cells that are activated by either extinction or reward conditioning. Approaches for accomplishing this type of manipulation are being developed, but it is not clear whether such approaches will work in these experimental paradigms. Nonetheless, recent work from Redondo et al. and Gore et al. both accomplished this type of manipulation in BLA, and these papers probably should have been discussed in the paper, as well as the recent paper from Namburi et al., which, though not using the type of manipulation used by Redondo and Gore, is also highly relevant.

3) Evaluation of physiological effects of optogenetic stimulation in anaesthetized rats.

I was disappointed that the examination of modulation in neural activity that could result from activation by optogenetic stimulation was done in anaesthetized rats. I understand why the authors may have done this for c-fos assays, but electrophysiology could have been performed in awake animals and short latency changes in activity could be observed in the absence of behavioral confound issues. Further, the authors could also inactivate the BLA while photoactivating terminals in NAc to test whether antidromic activation is playing a role. While I would not insist that the authors re-do all of these experiments, appropriate text needs to be added to the Discussion given that the findings may not hold in the absence of anaesthesia.

4) Accessibility of the manuscript.

I found the manuscript a difficult read. There is a lot of data, and it is often densely presented. Some of the figure legends are a page long, and many readers just won't engage them. Transitions between paragraphs often lack conceptual links. The Discussion reads in a somewhat rambling way, and parts of it are quite repetitive with the results. It takes many paragraphs for the authors to summarize their data rather having 1 or at most 2 succinct paragraphs followed by focused discussion. Again, I will not insist on changes here, but I think the authors would be well-served to revise the manuscript with these issues in mind; readers will find the work more accessible if they do so.

*Reviewer #3:*

This is an interesting and innovative report that shows that the activation of BLA-NAc circuit may underlie the facilitation effect on fear extinction caused by reward pairing. The data are important, and may provide a neural circuit mechanism for the counterconditioning therapy used to treat fear-related disorders. The topic has significant clinical relevance and is of great general interest. It is potentially suitable for publication in *eLife*. However, several issues need to be addressed or clarified before the full impact of these results can be determined.

In a first series of experiments, the authors performed double labeling experiments with retrograde tracer and cFos staining to show that the BLA-NAc circuit is recruited by fear extinction. However, I'm not fully convinced by the authors' implication that fear extinction activated extra NAc-projection neurons in BLA compared with the Fear group (Figure 1). This is an unfair comparison for the following reasons. First, the authors chose to perfuse the animals 1 hr after the final behavior session. cFos signals usually peaks at 1.5~2hr after the behavior onset. As fear recall session perhaps lasts much shorter than fear extinction (I could not find details for the behavior protocol. How long is the extinction session? How many trials does it consist? Such essential information is missing from the Methods.), it is likely that less cFos signals in the Fear group was simply due to a shorter expression time from the stimulus onset. Second, the fear extinction session consists many more trials than the fear recall session, so should naturally induce more signals. It will be useful to compare the percentage of double-labeled neurons relative to cFos+ neurons following fear recall and fear extinction. The authors showed 31% for the extinction group. Is the percentage lower for the recall group?

To rule out the antidromic effect on behavior, a standard way would be inactivating the upstream BLA region while optogenetically activating the terminals in NAc. If the effect still lasted, it would be much more convincing than recording or cfos counting in anesthetized animals. This is technically feasible, as demonstrated by the NAc inactivation experiment in this manuscript.

Anesthetized animals are not the most ideal if awake behaving animals can be used. For measuring of IL cFos after BLA-NAc stimulation, the authors should use awake animals, since that's the physiological condition under which behaviors are measured.

Have the authors analyzed the correlation between the level of extinction and that of reward conditioning at the individual level?

[Editors' note: further revisions were requested prior to acceptance, as described below.]

Thank you for submitting your article "Amygdala-ventral striatum circuit activation decreases long-term fear" for consideration by *eLife*. Your article has been reviewed by three peer reviewers, and the evaluation has been overseen by a Reviewing Editor and a Senior Editor. C Daniel Salzman (Reviewer #2) has agreed to reveal his identity.

The reviewers have discussed the reviews with one another and the Reviewing Editor has drafted this decision to help you prepare a revised submission.

All the reviewers thought that the authors have addressed many of the previous concerns by adding new data and revising the manuscript (please see below for individual comments). However, during discussion, the reviewers found that there are some concerns that need to be addressed before publication. Specifically, although reviewers 1 and 2 initially found the revisions satisfactory, after discussion with reviewer 3 and the reviewing editor, all of us concurred that the request for additional experiments was reasonable.

In the previous review, the reviewers were concerned that there is no evidence supporting the specificity of the BLA-NAc circuit recruitment during fear extinction and/or reward conditioning. The authors have addressed this concern by removing "specific" from the texts. In addition, it was pointed out, in the previous review, that only 5% of the BLA-NAc circuit was labeled by c-fos while this pathway was activated optogenetically without further specificity. Although the authors point out that ChR2 is expressed in only a fraction of BLA-NAc projections, this does not solve the issue that ~95% of stimulated neurons are not c-fos positive. Furthermore, in response to the reviewer #3's comments regarding c-fos experiments in anesthetized animals (which was not "essential" in the decision letter), the authors state that "it is unclear to us what additional information could be provided by the experiment suggested". We believe that performing c-fos experiments in awake animals is necessary to remedy the concern that the neuronal activity in anesthetized animals might not be normal.

Overall, the authors' responses to the above points were judged unsatisfactory. Whether the recruitment of BLA-NAc circuit has any predominant role in fear extinction remains unclear, and it is unclear whether optogenetic stimulation mimics physiological activities. The reviewers thought that these points are central to the present work and it is unlikely that these issues can be addressed simply by removing "specific". Ideally, the specificity of BLA-NAc circuit recruitment should be tested experimentally by showing that there is less c-fos activation of another projection after late extinction (which we believe can be done in 1 month). In addition, performing c-fos experiments in awake animals is necessary to support the authors' claims that ontogenetic stimulation did not cause activation of collaterals and that the c-fos expressions observed after optogenetic stimulation reflect normal circuit effects.

Overall, we strongly prefer that you perform the above experiments. Even if the authors decide not to perform these experiments, the above issues must be discussed more carefully and thoroughly so that the readers can appreciate the major potential caveats of the experiments and interpretations.

*Reviewer #1:*

The authors have adequately addressed my concerns.

*Reviewer #2:*

I think the authors have fully taken into account my previous concerns, and I now support publication.

*Reviewer #3:*

The authors have addressed many of the comments and the manuscript has been improved with a good amount of data. However, I 'm surprised that the authors insisted on using c-fos data in anesthetized animals as critical evidence for specific circuit activation.

It is well known that anesthesia strongly alters the excitation/inhibition balance in the brain. Some neurons which are activated under normal physiological conditions could be silenced by unnaturally strong inhibition under anesthetized state, and appear c-fos negative. On the other hand, if the anesthesia hits more heavily on a group of GABAergic neurons that disinhibits other neurons, the latter neurons may appear falsely c-fos positive. Therefore, interpretation of c-fos data from anesthetized animals could be very misleading.

As the author was not able to show that BLA-NAc is selectively activated by extinction, the NAc terminal stimulation experiment became critical in addressing the specific role of the pathway in extinction. The authors took the absence of c-fos signals in YFP+ BLA neurons in anesthetized animals as the only evidence to rule out antidromic effect by the terminal stimulation. While I understand there might be technical issues involved in inactivating upstream BLA region, I do not see why the authors should not perform the c-fos experiment in awake animals, which is much more physiologically-relevant.

Similarly, activation of IL is an important mechanism the authors proposed to explain why BLA-NAc stimulation facilitates extinction. I am disappointed that this was again only supported by results in anesthetized animals. c-fos experiments in awake animals are totally feasible and not much more tedious than in anesthetized animals. The authors should repeat these two sets of experiments in awake animals.

---

## [Author Response]

*Essential revisions:*

*1) Specificity of the recruitment of the BLA-NAc circuit during extinction and/or reward conditioning. It is not clear whether the increase in c-fos labeled BLA cells occurs preferentially in the NAc projecting cells or generally in the BLA. If this is occurring by chance, then the evidence for specific recruitment of the BLA-NAc pathway would not be provided. This point needs to be explicitly analyzed.*

We did not intend to claim that the activation of the BLA-NAc pathway is selective to extinction. We believe this issue may have arisen because we used the term specific to describe the anatomical and temporal specificity of our manipulations. It is clear that the double-labeling of cFos and CTB (Figure 1), indicating activation of the BLA-NAc pathway, strongly parallels the elevated levels of cFos observed in all cells of the BLA (Figure 1). We have modified the language of the text in our manuscript to clarify what we mean by specific each time it is used, and we also have almost eliminated our use of this term.

To directly address the issue of behavioral specificity, we conducted an additional experiment (Figure 1—figure supplement 2). We examined the activation of this pathway by fear conditioning (*Fear Cond* group) and the activation of this pathway by lower levels of extinction training (*Short ext* group). We found that the BLA-NAc pathway was activated by fear learning. However, more importantly, and more relevant to our claims, we found that one session of extinction training recruited this pathway to a lesser degree than two sessions of extinction training. Critically, we also found that there is a strong, negative association between the extent to which the BLA-NAc pathway is activated and the fear expressed (Figure 1—figure supplement 2). We have added a paragraph to the Discussion to consider why both fear conditioning and fear extinction may activate the same BLA-NAc circuit.

*2) Data exclusions. It was noted that the authors excluded some samples/data, which might have biased some of the data analyses. The authors need to clarify this.*

We have modified the Methods to explicitly disclose the number of rats excluded for these behavioral exclusion criteria. Of particular concern, Reviewer 1 noted that we excluded rats from the *Ext-RC* and *RC* groups that failed to display any indication of learning (either a failure to discriminate between the tone paired with sucrose and the tone never paired with sucrose, or a failure to nose-poke). The number of rats excluded for these reasons was very low (n=6). As requested by the Reviewer, we show that their behavior during fear conditioning and the first two extinction sessions did not differ from the other groups (see below). Thus, there is no data to support the claim that the data analysis is biased by the exclusions.

*3) Potential non-physiological effects of optogenetic manipulation. The data suggests that 95% of the neurons activated by optogenetic manipulation were not activated by fear conditioning and extinction. The authors need to consider whether the effects they observe are non-physiological effects.*

We have added a paragraph to our Discussion to thoughtfully consider this point. While we acknowledge that optogenetic stimulation may produce supraphysiological effects, it is unlikely that the effects we observe are nonphysiological. The criticism that 95% of the neurons activated by optogenetic manipulation were not activated by fear conditioning+extinction rests on a faulty assumption that optogenetic stimulation activates 100% of the BLA-NAc circuit. However, it should be noted that viral expression of opsins in the BLA targeted only a subset of BLA cells, and thus only a portion of the BLA cells that project to the NAc. It is also important to note that our CTB infusions targeted only a small portion of the NAc, and thus identified only a subset of the BLA cells that project to this area. It is likely that our optogenetic manipulation likely targeted a portion of cells not activated by fear conditioning and extinction, but not nearly to the dramatic degree indicated here (95%). In addition, the magnitude of enhanced extinction persistence seen with stimulation (Figure 5) is highly similar to the magnitude of enhanced extinction persistence seen with reward conditioning (Figure 2), and these conditions that lead to greater persistence of extinction memory are associated with stronger activation of the IL but not PL (Figure 6). Thus, the effects of optogenetic stimulation strongly parallel the effects induced by natural behavior.

As suggested by Reviewer 2, we have added text to the Discussion to consider how behavior-specific expression of opsins could be used to better restrict stimulation to the portion of the BLA-NAc circuit that is relevant to extinction. We greatly appreciate this suggestion.

*4) One of the reviewers raised various concerns in the experimental design and data analysis regarding the conclusion that the BLA-NAc circuit is recruited by fear extinction. See below for the details.*

It is unclear whether this is a reference to Reviewer 1, point 3, or Reviewer 2, point 1. We have made extensive revisions in response to both comments. Please see below for the details.

*Reviewer #1:*

*1) For Figure 1, it is not clear whether the increase in cfos labeled BLA cells occurs preferentially in the NAc projecting cells or generally in the BLA. While the authors mention in the fifth paragraph of the subsection A BLA-NAc circuit is recruited by fear extinction that the a third of the cfos labeled cells are NAc projecting, they need to use a more rigorous statistical test to demonstrate that recruitment of this pathway is actually selective.*

This is an important point, and we appreciate the Reviewer asking us to provide additional clarification. It was never our intention to claim that the activation of the BLA-NAc pathway is selective to extinction. While we did use the term specific to describe the recruitment of the BLA-NAc pathway during this behavior, we were using this to describe the anatomical and temporal specificity of our manipulations. Indeed, it is clear that the double-labeling of cFos and CTB (Figure 1), showing activation of the BLA-NAc pathway, strongly parallels the elevated levels of cFos observed in all cells of the BLA (Figure 1). We have modified the language of the text in our manuscript to clarify what we mean by specific, and we also greatly reduced our use of this term.

To better address this issue, we conducted an additional experiment (Figure 1—figure supplement 2). We examined the activation of this pathway by fear conditioning (*Fear Cond* group), and the activation of this pathway by lower levels of extinction training (*Short ext* group). We found that the BLA-NAc pathway was activated by fear learning. However, more importantly, and more relevant to our claims, we found that one session of extinction training recruited this pathway to a lesser degree than two sessions of extinction training. Critically, we also found that there is a strong, negative association between the extent to which the BLA-NAc pathway is activated and the fear expressed (Figure 1—figure supplement 2).

It is important to note that brain circuits are rarely, if ever, behaviorally specific. Thus, the finding that the BLA-NAc pathway is recruited by fear conditioning does not detract from its role in fear extinction. Our data demonstrate clearly that optogenetic activation of this pathway enhances the persistence of extinction memory.

We have added a paragraph to the Discussion (third paragraph) to consider the surprising observation that the BLA-NAc pathway is activated by fear conditioning. In this paragraph, we explicitly state that the BLA-NAc circuit is not selective for fear extinction and we offer two possible explanations for why this circuit may be activated by both behaviors.

*2) Related to the counterconditioning data presented in Figure 2, the authors mention in the Methods that "Rats that never displayed reward-seeking (failure to nose-poke) or discriminative reward conditioning (failure to nose-poke significantly more or faster to the CS paired with reward than to the neutral CS) were excluded from all analyses." This is potentially a major issue as they are selectively removing animals from one group. This could bias their results in the counterconditioning group such that the animals in this group have reduced fear specifically because these animals were removed and not because the counterconditioning enhanced extinction memory retention. For example, it is possible that animals that couldn't learn reward seeking had higher fear levels or more spontaneous recovery of extinction. The authors need to do something to address this problem. One possibility is to identify animals that would be excluded and then do extinction training and compare these animals that would have been excluded to a normal group of extinction animals to see if there are differences in spontaneous recovery. If not, then there is not a problem with the exclusion criteria.*

This is an important issue to address. We now report the numbers of these rats excluded for the three experiments affected by this criterion (subsection Reward conditioning, second paragraph). As suggested by the Reviewer, we examined their freezing behavior on Days 1-3 of the experiment, and did not observe any difference in fear acquisition or extinction during this period. These data are shown in Figure 7. It is clear that the freezing behavior on Days 1-3 does not differ between the excluded rats (Ext-RC and did not learn RC) and the included rats (Ext-RC and Learned RC) (Mann-Whitney, p = 0.46).

Author response image 1.**DOI:**
http://dx.doi.org/10.7554/eLife.12669.020

*A) Related to the above comment, they should state somewhere how many animals were excluded. Also, for the data in Figure 2, the authors should clarify whether this does or does not include excluded animals.*

The numbers of rats excluded and the reasons for exclusion are now specified in the second paragraph of the subsection Reward conditioning. We now clarify in the Methods (subsection Statistical methods, second paragraph) that no data from excluded animals were included in any graph or analysis.

*3) The authors should clarify in the Methods section whether the exact same box was used (including the availability of all apparatuses in box except reward) for both ext-ext and ext-RC groups? This is important as differences in the box could produce differences in extinction.*

This information is now provided in the Methods. To summarize, the reward port was present as an environmental cue for all Context B sessions for all groups in experiments where reward counterconditioning was used. The only difference was whether reward was delivered through the port during the session.

*4) There are a number of issues with data presented in Figure 3:*

*A) It is not clear why there is an inset to compare Ext-RC and RC groups. Ext-Ext, Ext-RC and RC groups should be in a single graph with appropriate statistical comparisons.*

We used insets simply to highlight the relevant comparisons, and also draw attention to the critical comparison of interest (Ext-Ext vs. Ext-RC). The appropriate statistical comparisons are reported in the Results and figure caption.

*B) A negative control group should be included here to determine whether there is an increase above baseline in fos activation. One of the control groups used in Figure 1 would be sufficient.*

The Reviewer is questioning whether the cFos activation in the Ext-Ext group is above baseline (the Nave group). This comparison was previously included in Figure 1, but has also been examined in Figure 1—figure supplement 2 of the revised manuscript. The expression of cFos in the BLA of Nave rats is extremely low and the expression of cFos cells in the *Ext-Ext* group is well above baseline.

*C) For the BLA data in 3B and C, representative images should be shown somewhere.*

We have added representative images as Figure 3 (bottom panel).

*D) One possibility is that extinction (or reward) preferentially activates NAc projectors early in learning and that the differences you see between extinction and reward learning are due to that and not because the extinction and reward are summating to activate more cells. It would ideal to run another group where tissue is collected after 1 day of extinction to match the single day of reward learning.*

Whether NAc-projecting BLA cells are preferentially activated by early or late learning is an important question. As requested, we have added a new experiment presented in Figure 1—figure supplement 2 to address this point. It is clear from this new data that NAc-projecting BLA cells are activated to a greater extent in late extinction as compared to early extinction. Thus, it is highly unlikely that the greater activation of the BLA-NAc pathway in the Ext-RC or RC groups arises because these groups were sacrificed after the first day of learning the reward task.

*E) Also need negative control for the NAc Nr4a3 data in Figure 3 as it is not clear whether there is a significant Nr4a3 activation above baseline in the ext-ext group.*

This data has been added as Figure 3—figure supplement 3. The software that we used for imaging in Figure 3 was replaced with an updated version with substantial differences. Because we could not image using the same parameters in Figure 3, we could not add data from a Nave group to Figure 3. We performed new labeling of tissue from Nave, Ext-Ext and Ext-RC groups.

*5) For the data presented in Figure 5, they should show supplemental data with raw% freezing levels on day 2 and day 60 in supplement as they do for the data in Figure 2.*

We apologize for the omission of some of the raw data. The raw % freezing for Day 2 was depicted in Figure 5 of our original submission and it is also depicted in this panel in the revised manuscript. The raw % freezing for Day 60 is now shown in Figure 5—figure supplement 2. In addition, we have confirmed that all raw freezing levels are depicted in graphs labeled as % Freezing, whereas our measure of spontaneous recovery at remote time points after extinction training is shown in graphs labeled % Fear memory. We have revised our Methods (subsection Pavlovian fear conditioning, last paragraph) to clearly indicate this distinction.

*6) Though not absolutely necessary, inhibition of BLA-NAc terminals during extinction would be ideal to directly test whether the BLA-NAc projection participates causally in generating extinction. As it stands they present correlative evidence that the circuit is engaged, but the paper would much more compelling if they could make a causal link.*

We fully agree that a causal link would be compelling. We present evidence that the enhanced recruitment of the BLA-NAc pathway during extinction facilitates long term fear extinction. However, we do not claim the necessity of this pathway for fear extinction learning.

*Reviewer #2:*

*1) Specificity of the recruitment of the BLA-NAc circuit during extinction and/or reward conditioning.*

*The authors' data largely rely on an observed increase following extinction or reward conditioning in double labeled cells in BLA using c-fos and CTB+ to mark activated cells that project to NAc. However, the authors do not address whether this increase in double labeling could be occurring by chance as a result of simply increasing the number of c-fos positive cells in BLA. In other words, increased double labeling would be expected simply by virtue of increasing the number of c-fos positive cells. If this is occurring by chance, then the evidence for specific recruitment of the BLA-NAc pathway would not be provided. Moreover, the statement that the proportion of activated BLA neurons that project to NAc increases with reward conditioning can't be supported without such statistical testing.*

This is an important point also noted by Reviewer 1. It was never our intention to claim that the activation of the BLA-NAc pathway is selective to extinction. While we did use the term specific to describe the recruitment of the BLA-NAc pathway during this behavior, we were using this to describe the anatomical and temporal specificity of our manipulations. Indeed, it is clear that the double-labeling of cFos and CTB (Figure 1), showing activation of the BLA-NAc pathway, strongly parallels the elevated levels of cFos observed in all cells of the BLA (Figure 1). We have modified the language of the text in our manuscript to clarify what we mean by specific, and we also almost eliminated our use of this term.

To better address this issue, we conducted an additional experiment (Figure 1—figure supplement 2). We examined the activation of this pathway by fear conditioning (*Fear Cond* group), and the activation of this pathway by lower levels of extinction training (*Short-term ext* group). We found that the BLA-NAc pathway was activated by fear learning. However, more importantly, and more relevant to our claims, we found that one session of extinction training recruited this pathway to a lesser degree than two sessions of extinction training. Critically, we also found that there is a strong, negative association between the extent to which the BLA-NAc pathway is activated and the fear expressed (Figure 1—figure supplement 2).

It is important to note that brain circuits are rarely, if ever, behaviorally specific. Thus, the finding that the BLA-NAc pathway is recruited by fear conditioning does not detract from its role in fear extinction. Our data demonstrate clearly that optogenetic activation of this pathway enhances the persistence of extinction memory.

*2) Potential non-physiological effects of optogenetic manipulation.*

*The authors report that only 5% of the NAc-projecting BLA neurons were double-labeled by c-fos and CTB+. This implies that 95% of the neurons activated by optogenetic manipulation were not activated by fear conditioning and extinction.*

This is not an accurate statement. While it is true that we found that 5% of the NAc-projecting BLA cells were double-labeled, this is an underestimate of this population. The CTB infusions target only a small portion of the NAc, and thus identify only a small subset of the BLA cells that project to this area. In addition, viral expression of the opsin in the BLA targets only a portion of the BLA, and thus only a portion of the BLA cells that project to the NAc. It is correct that our optogenetic manipulation likely targeted a substantial portion of cells not activated by fear conditioning and extinction, but not nearly to the dramatic degree indicated here (95%).

*The authors need to consider whether the effects they observe are non-physiological effects (essentially activating neurons not normally activated during the relevant behavior) given that they don't know much about the response properties of most of the neurons they are activating optogenetically. This type of issue applies to many (if not most) studies employing optogenetic methodology, but it is one that I think we all need to worry about more.*

We fully agree with these statements and have added a new paragraph to the Discussion to acknowledge this shortcoming (twelfth).

*A missing link in this paper is the specific activation only of those BLA cells that are activated by either extinction or reward conditioning. Approaches for accomplishing this type of manipulation are being developed, but it is not clear whether such approaches will work in these experimental paradigms. Nonetheless, recent work from Redondo et al. and Gore et al. both accomplished this type of manipulation in BLA, and these papers probably should have been discussed in the paper, as well as the recent paper from Namburi et al., which, though not using the type of manipulation used by Redondo and Gore, is also highly relevant.*

We agree with the Reviewer and have added citations for these three references in the Discussion. In addition, we now acknowledge that activity-dependent expression of opsins is an appealing approach for overcoming the non-physiological effects of stimulation.

*3) Evaluation of physiological effects of optogenetic stimulation in anaesthetized rats.*

*I was disappointed that the examination of modulation in neural activity that could result from activation by optogenetic stimulation was done in anaesthetized rats. I understand why the authors may have done this for c-fos assays, but electrophysiology could have been performed in awake animals and short latency changes in activity could be observed in the absence of behavioral confound issues. Further, the authors could also inactivate the BLA while photoactivating terminals in NAc to test whether antidromic activation is playing a role. While I would not insist that the authors re-do all of these experiments, appropriate text needs to be added to the Discussion given that the findings may not hold in the absence of anaesthesia.*

We have now added a paragraph to the Discussion (thirteenth paragraph) to explicitly acknowledge that quantitative findings may not hold in the absence of anesthesia. We also provide citations showing that cFos in the NAc and IL is not impacted by isoflurane anesthesia in other studies.

*4) Accessibility of the manuscript.*

*I found the manuscript a difficult read. There is a lot of data, and it is often densely presented. Some of the figure legends are a page long, and many readers just won't engage them. Transitions between paragraphs often lack conceptual links. The Discussion reads in a somewhat rambling way, and parts of it are quite repetitive with the results. It takes many paragraphs for the authors to summarize their data rather having 1 or at most 2 succinct paragraphs followed by focused discussion. Again, I will not insist on changes here, but I think the authors would be well-served to revise the manuscript with these issues in mind; readers will find the work more accessible if they do so.*

We thank the Reviewer for these comments; we appreciate the difficulty of presenting substantial quantities of data in an approachable manner. We have thoroughly edited our figure captions to shorten them by removing all redundant details. We have revised both our Results and Discussion sections to be more concise, while also adding in the new data and discussion points requested by all three Reviewers. We have added a brief summary at the start of the Discussion and removed redundant summary statements from the remainder of the Discussion.

*Reviewer #3:*

*In a first series of experiments, the authors performed double labeling experiments with retrograde tracer and cFos staining to show that the BLA-NAc circuit is recruited by fear extinction. However, I'm not fully convinced by the authors' implication that fear extinction activated extra NAc-projection neurons in BLA compared with the Fear group (Figure 1). This is an unfair comparison for the following reasons. First, the authors chose to perfuse the animals 1 hr after the final behavior session. cFos signals usually peaks at 1.5~2hr after the behavior onset.*

We performed an additional experiment to test this. We trained additional rats with two extinction sessions (as per Figure 1) and harvested the brains either 1 or 1.5 hr following the last extinction session. As shown in Figure 1—figure supplement 1, cFos protein expression in the BLA was significantly lower at 1.5 hr post-extinction than at 1 hr post-extinction.

It should also be noted that while reward-induced cFos mRNA expression in the BLA peaks 1.5 hr after behavioral onset and declines thereafter, cFos protein expression in the BLA reaches its peak at 1.5 hr after behavioral onset and remains at its asymptotic level for at least 30 min beyond this (see Xiu et al., Nat. Neurosci., 2014). Because our extinction sessions last approximately one hour and we sacrificed the rats one hour following this, we examined cFos expression at roughly two hours after behavioral onset, a time point consistent with the expected peak of cFos protein expression in the BLA.

*(I could not find details for the behavior protocol. How long is the extinction session? How many trials does it consist? Such essential information is missing from the Methods.),*

We apologize that this information was not included in the Methods. We have revised the text to explicitly state the length and the total number of trials in the fear conditioning, extinction, and recall sessions (subsection Pavlovian fear conditioning, last paragraph).

*As fear recall session perhaps lasts much shorter than fear extinction, it is likely that less cFos signals in the Fear group was simply due to a shorter expression time from the stimulus onset.*

The Reviewer has made an astute observation, noting that the lower levels of activation of the BLA-NAc pathway in the *Fear recall* group compared to the *Long ext* group (formerly the *Ext* group) could be because these groups spend different amounts of time in the chamber. To address this, we added a new experiment (Figure 1—figure supplement 2) in which we compared activation of the BLA-NAc pathway in a group of rats that received one day of extinction training (*Short ext* group) to activation of this pathway in a group of rats that received two days of extinction training (*Long ext* group). Importantly, the brains from both groups of animals were collected after the same behavioral experience (extinction lasting roughly 30 min); they differed only in whether they also experienced an extinction session the day before. We found that one session of extinction training recruited the BLA-NAc pathway to a lesser degree than two sessions of extinction training. Critically, we also found that there is a strong, negative association between the extent to which the BLA-NAc pathway is activated and the fear expressed (Figure 1—figure supplement 2).

*Second, the fear extinction session consists many more trials than the fear recall session, so should naturally induce more signals. It will be useful to compare the percentage of double-labeled neurons relative to cFos+ neurons following fear recall and fear extinction. The authors showed 31% for the extinction group. Is the percentage lower for the recall group?*

As we note above, our new data show that session length does not determine the degree of activation of the BLA-NAc pathway. The Reviewer suggests that we could compare the percentage of the BLA-NAc pathway that is activated under different conditions. It is 31% for the *Long ext* group and 37% for the *Fear recall* group. In general, we find that any condition that activates the BLA-NAc pathway activates ~1/3 of the pathway. We do not understand why this occurs. It may be related to the fact that CTB infused into the NAc only labels a portion of the pathway. Alternatively, it may be that the number of cells activated by learning in the BLA is the result of a conserved, competitive process (Han et al., Science, 2007), and that the downstream impact in the NAc reflects this conserved process.

*To rule out the antidromic effect on behavior, a standard way would be inactivating the upstream BLA region while optogenetically activating the terminals in NAc. If the effect still lasted, it would be much more convincing than recording or cfos counting in anesthetized animals. This is technically feasible, as demonstrated by the NAc inactivation experiment in this manuscript.*

We agree that this is technically feasible, but we believe that this labor-intensive experiment is unlikely to yield interpretable results. We show that optogenetic activation of BLA terminals in NAc deepens the persistence of fear extinction learning. Temporary inactivation of the BLA strongly impairs extinction learning and long-term extinction memory (Sierra-Mercado et al., Neuropsychopharmacology, 2011). If inactivation of the BLA during extinction learning *without* optogenetic stimulation impairs long-term extinction memory, then one cannot use this manipulation to determine whether the modulatory effect of optogenetic stimulation is eliminated. It is for this reason that we performed the experiment to characterize the impact of BLA terminal stimulation in the NAc in anesthetized animals. If substantial antidromic activation of the BLA occurred, it should have induced cFos expression in the BLA cells expressing the opsin (eYFP+), but we did not observe this (Figure 6). This provides strong evidence that there was minimal antidromic activation of the BLA when stimulating BLA terminals in the NAc.

*Anesthetized animals are not the most ideal if awake behaving animals can be used. For measuring of IL cFos after BLA-NAc stimulation, the authors should use awake animals, since that's the physiological condition under which behaviors are measured.*

It is unclear to use what additional information could be provided by the experiment suggested. We show clearly, in anesthetized rats, that optogenetic stimulation of BLA terminals in the NAc leads to activation of the IL (Figure 6). We make no claims that this is a quantitative representation of what occurs in awake, behaving rats. We also show, in awake, behaving rats, that the IL is more strongly recruited by extinction training followed by reward conditioning than extinction training alone (Figure 6). It is this latter point which is the most important and relevant to our claims.

*Have the authors analyzed the correlation between the level of extinction and that of reward conditioning at the individual level?*

We have performed this analysis, correlating the% Freezing on Day 55 with the latency to nose-poke on Day 5 (after the second day of reward conditioning), across individual animals. As can be seen in Figure 8, there is no significant correlation (R^2^ = 0.035). This may be because most rats respond to the CS+ with similar nose-poke latencies when the task is acquired.

Author response image 2.**DOI:**
http://dx.doi.org/10.7554/eLife.12669.021

[Editors' note: further revisions were requested prior to acceptance, as described below.]

In the previous review, the reviewers were concerned that there is no evidence supporting the specificity of the BLA-NAc circuit recruitment during fear extinction and/or reward conditioning. The authors have addressed this concern by removing "specific" from the texts.

Overall, the authors' responses to the above points were judged unsatisfactory. Whether the recruitment of BLA-NAc circuit has any predominant role in fear extinction remains unclear, and it is unclear whether optogenetic stimulation mimics physiological activities. The reviewers thought that these points are central to the present work and it is unlikely that these issues can be addressed simply by removing "specific". Ideally, the specificity of BLA-NAc circuit recruitment should be tested experimentally by showing that there is less c-fos activation of another projection after late extinction.

This criticism pertains to a fundamental issue of whether the BLA-NAc recruitment by extinction is *specific*. There are several definitions of specificity, and it is incorrect to characterize our response to this criticism as simply removing the word specific from the text.

Specificity may refer to behavioral specificity. We previously performed an additional experiment to show that activation of the BLA-NAc circuit was *not*specific to fear extinction; fear conditioning produced a similar activation of this circuit.

Specificity may also refer to anatomical specificity, meaning that only certain outputs of any given brain region are recruited by behavior. It has been argued by the Reviewers that the activation of the BLA-NAc pathway that we reported is not anatomically specific because it simply mirrors overall activation [measured by immediate early gene (IEG) expression] within BLA after extinction learning. However, to argue that activation measured by IEG expression is not related to behavior requires that IEG expression be expressed randomly. However, IEG expression during memory encoding and retrieval forms the basis for tag and manipulate strategies to identify engrams, and is thus a reliable marker for cells that participate in memory formation or retrieval (Josselyn, Keler, Frankland, Nat. Neurosci. Rev., 2015). Our claim is that lengthy extinction training activates a greater proportion of the BLA than is observed in nave rats or rats in the early stages of extinction, including BLA cells that project to the NAc.

It was argued that anatomical specificity should be shown by demonstrating that at least one other BLA projection is not activated by extinction learning. While we agree that the proposed experiment is an interesting idea, this is not a simple experiment. It is not obvious what BLA projection would exhibit little to no cFos activity after fear extinction (particularly because cFos may mark both synaptic strengthening and synaptic weakening), and thus multiple projections would need to be tested. This systematic analysis of different projections from the BLA would be very informative for the field and would, on its own, provide data for a single publication.

The Reviewing Editor suggested that a promising pathway to show the absence of activation by extinction learning might be the BLA-CeM because Namburi et al. (2015) have argued that this pathway is specifically activated by fear learning, not reward learning. It is important to acknowledge two specific considerations regarding this point. First, fear extinction sessions involve both the expression of previously acquired fear as well as the acquisition of extinction. Thus, it is highly likely that the BLA-CeM pathway is activated during fear extinction. Second, although the Namburi et al. paper is often cited as a demonstration of anatomical specificity, this is a mis-interpretation of the findings. The authors delivered retrobeads to either the CeM or the NAc to visualize BLA cells that projected to these brain regions. The findings reported by Namburi et al. hinge on the argument that BLA cells project to exclusively to the NAc or the CeM. However, the mutual exclusivity of these BLA projections was not quantified or demonstrated. It is apparent from Figure 9 (modified from Figure 1B in Namburi et al.) that a substantial portion of BLA cells project to both the CeM and NAc (shown by orange arrows which indicate cell bodies which contain retrobeads from both CeM and NAc). In this regard, for all of the experiments in Namburi et al., a substantial proportion of NAc projectors or CeM projectors actually projected to both targets. Thus, this paper should not be used to support claims of anatomical specificity.

Author response image 3.**DOI:**
http://dx.doi.org/10.7554/eLife.12669.022

It is essential to recognize that our claim that extinction learning recruits a BLA-NAc circuit is supported by multiple findings. For example, the recruitment of this pathway by extinction is strongly related to the degree of extinction (greater pathway activation = less fear; Figure 1—figure supplement 2) and we had already shown that selective activation of the BLA-NAc projection deepens extinction learning (Figure 5). We also observe that activation of the NAc by extinction training (measured via Nr4a3 expression in NAc) is strongly related to the degree of extinction (Figure 10; greater NAc activation = less fear). If activation of the BLA-NAc pathway was truly the result of chance, then one would not expect to observe a relationship between the magnitude of extinction learning and activation within this pathway. We have added text to consider this point in the Discussion (second paragraph).

Author response image 4.**DOI:**
http://dx.doi.org/10.7554/eLife.12669.023

*In addition, it was pointed out, in the previous review, that only 5% of the BLA-NAc circuit was labeled by c-fos while this pathway was activated optogenetically without further specificity. Although the authors point out that ChR2 is expressed in only a fraction of BLA-NAc projections, this does not solve the issue that ~95% of stimulated neurons are not c-fos positive.*

An additional concern was raised because we report that extinction learning produces activation of only 5% of the BLA-NAc pathway. However, there are several additional points which are not being considered when making this assertion. First, as we noted in the text, nearly one third of all cFos induced by extinction training is expressed in BLA cells that project to the NAc. This is a large proportion of the cFos induced by extinction training. Second, the projection from the BLA to the NAc is substantial. We have added a new analysis showing that ~27% of the cells in the BLA regions we analyzed have projections to the NAc, a finding consistent with previous studies (McDonald, 1991). We have modified the text of the Results (subsection A BLA-NAc circuit is recruited by fear extinction, fourth paragraph) to include this information.

Regardless, we do agree that optogenetic stimulation cannot precisely model physiological activities under the experimental conditions used, as we are randomly infecting and stimulating BLA neurons with terminals in the NAc. We have provided additional text to more deeply consider this caveat in the Discussion of the paper (fourteenth paragraph).

*Furthermore, in response to the reviewer #3's comments regarding c-fos experiments in anesthetized animals (which was not "essential" in the decision letter), the authors state that "it is unclear to us what additional information could be provided by the experiment suggested". We believe that performing c-fos experiments in awake animals is necessary to remedy the concern that the neuronal activity in anesthetized animals might not be normal.*

We agree that doing stimulations in awake and anesthetized animals is very different, but both approaches are hampered by different caveats:

In the case of anesthetized recordings, we agree with Reviewer 3 that the isoflurane anesthesia may alter the balance of excitation and inhibition, which could result in false positive cFos (due to disinhibition) or lack of normally activated cFos neurons (due to extra inhibition). However, previous studies looking at cFos expression in the amygdala, infralimbic cortex, nucleus accumbens and other brain areas have not found a significant difference between animals exposed to a room versus animals that undergo isoflurane anesthesia (Smith et al., Neurosci., 2016). Additionally, Figure 6 shows that anesthesia does not alter cFos numbers in the BLA as compared to Nave awake rats, indicating that, in the absence of stimuli, false positive cFos (due to disinhibition) is highly unlikely. It remains possible that anesthesia could increase activity of inhibitory interneurons in the BLA and therefore mask potential antidromic activation of BLA neurons. We have modified our Discussion to acknowledge the important caveats in experiments with anesthetized preparations (twelfth and fifteenth paragraphs).

It is also important to note that other studies have shown that stimulation of the nucleus accumbens in awake animals activates the IL (Vassoler et al., J. Neurosci., 2013), just as we report here in anesthetized animals. Thus, this provides further evidence that our stimulation paradigm produces physiologically relevant effects in circuits known to regulate fear extinction. It is also important to note that our stimulation produced activation of IL only in the IL contralateral to the hemisphere of stimulation, consistent with the finding that the dominant projection of the NAc to the IL is contralateral. It is also important to note that stimulation produced a much smaller activation within PL, consistent with previous findings that the NAc projections to the IL are much greater than to the PL. Finally, we also reported that extinction training activated the IL, but not the PL. We also agree with the comments of Reviewer 3 and now we consider these as potential caveats in the interpretation of the results from the optogenetic stimulation experiments in anesthetized rats. But we also note that there were strong parallels between what we observed in anesthetized rats and what we observed in awake, behaving rats.

*Reviewer #3:*

*The authors have addressed many of the comments and the manuscript has been improved with a good amount of data. However, I 'm surprised that the authors insisted on using c-fos data in anesthetized animals as critical evidence for specific circuit activation.*

*It is well known that anesthesia strongly alters the excitation/inhibition balance in the brain. Some neurons which are activated under normal physiological conditions could be silenced by unnaturally strong inhibition under anesthetized state, and appear c-fos negative. On the other hand, if the anesthesia hits more heavily on a group of GABAergic neurons that disinhibits other neurons, the latter neurons may appear falsely c-fos positive. Therefore, interpretation of c-fos data from anesthetized animals could be very misleading.*

*As the author was not able to show that BLA-NAc is selectively activated by extinction, the NAc terminal stimulation experiment became critical in addressing the specific role of the pathway in extinction. The authors took the absence of c-fos signals in YFP+ BLA neurons in anesthetized animals as the only evidence to rule out antidromic effect by the terminal stimulation. While I understand there might be technical issues involved in inactivating upstream BLA region, I do not see why the authors should not perform the c-fos experiment in awake animals, which is much more physiologically-relevant.*

As the editor noted above, this experiment (BLA inactivation during stimulation) was not "essential" in the previous decision letter; because it was a non-essential revision and because of the lengthy nature of these experiments, we decided not to pursue this experiment after the first review. We performed the experiments in anesthetized animals because we believed that the anesthesia was likely to interfere less with the cFos induced by optogenetic stimulation in different brain areas than behavior, which itself induces cFos expression in multiple brain circuits. We fully agree with the Reviewer that these are important concerns. We have thoroughly revised the Discussion to provide a fuller and more balanced perspective on the caveats of anesthetized preparations (twelfth and fifteenth paragraphs).

*Similarly, activation of IL is an important mechanism the authors proposed to explain why BLA-NAc stimulation facilitates extinction. I am disappointed that this was again only supported by results in anesthetized animals. c-fos experiments in awake animals are totally feasible and not much more tedious than in anesthetized animals. The authors should repeat these two sets of experiments in awake animals.*

We fully understand the reviewers concerns. However, again we would like to point out that studies looking at cFos expression in the amygdala, infralimbic cortex and nucleus accumbens have not found a significant difference between animals exposed to the anesthesia room versus animals that undergo isoflurane anesthesia (Smith et al., Neurosci., 2016). Therefore, while we consider the possibility that the increase in cFos+ cells in the IL after BLA-NAc stimulation could be due to false positive cFos (due to disinhibition), we also note that 1) other studies have shown that stimulation of the nucleus accumbens in awake animals activates the IL (Vassoler et al., J. Neurosci., 2013), just as we report here in anesthetized animals, 2) our stimulation produced activation of IL only in the IL contralateral to the hemisphere of stimulation, consistent with the finding that the dominant projection of the NAc to the IL is contralateral, 3) stimulation produced a much smaller activation within PL, consistent with previous findings that the NAc projections to the IL are much greater than to the PL, and 4) extinction training activated the IL, but not the PL. Thus, there were strong parallels between what we observed in anesthetized rats and what we observed in awake, behaving rats. Nonetheless, we have revised the Discussion to fully acknowledge the shortcomings of using anesthesia (twelfth and fifteenth paragraphs).